# Transient boosting of action potential backpropagation for few-shot temporal pattern learning

Gaston Sivori, Tomoki Fukai [ID]*

Okinawa Institute of Science and Technology, Onna-son, Okinawa, Japan

* tomoki.fukai@oist.jp

## Abstract

One of the remarkable properties of cortical neurons is their innate ability to detect spike patterns in continuous but noisy information streams. As animals learn behaviorally relevant information quickly, pattern detection by neurons should also be rapid. However, how pattern-selective neuronal responses rapidly develop, tune, and remain robust has not been fully understood. Here, we propose a biologically plausible synaptic plasticity rule to learn patterned synaptic inputs rapidly. Our rule facilitates intracellular "self-supervised" learning of intermittently and repeatedly co-activated presynaptic-neuron communities, as in a machine-learning-based rule derived previously for temporal pattern segmentation. Importantly, this model proposes that a spike-triggered transient increase in somatodendritic coupling dramatically boosts the crediting of synapses responsible for the learned response. This boosting effect is essential for rapid pattern learning of single neurons at a high signal-to-noise ratio. Moreover, we demonstrate in recurrent networks how the rule recruits pre-configured cell assemblies for even faster, few-shot learning of multiple patterns. Our results illuminate the self-supervisory role of backpropagating action potentials for rapid pattern learning.

## Author summary

To what extent neural computations underlying brain computation and AI share the principles and mechanisms of computations is poorly understood. Many biological features of the brain are ignored in the state-of-the-art AI machines, making the artificial neural networks oversimplified imitations of biological neural circuits. Recent experimental and computational studies suggest that the dendrites enable single biological neurons to learn and perform very complex computations. Therefore, neural circuits in the brain may be regarded as networks of highly sophisticated computational units, a view that has been largely ignored in studies

**Data availability statement:** All numerical datasets necessary to replicate our results can be generated with the software code referenced below. All codes were written using standard libraries with Julia programming language version 1.8, Python version 3. Example program codes used for numerical simulations and data analysis are available at

https://github.com/gsivori/transientboost/ (10.5281/zenodo.14792253).

**Funding:** T.F. receives KAKENHI JP23H05476 from the Japan Society for the Promotion of Science (JSPS: https://www.jsps.go.jp/english/). The funders had no role in study design, data collection and analysis, decision to publish, or preparation of the manuscript.

**Competing interests:** The authors have declared that no competing interests exist.

of artificial intelligence. In this study, we have proposed a biologically plausible mechanism of the unsupervised learning of salient input patterns, a single-cell computation hypothesized in many previous models of cognitive functions. To our surprise, the biological mechanism enables the neuron model to detect salient input patterns much more rapidly than a machine-learning-based neuron model. Furthermore, this rapid learning is further accelerated when the neuron model forms cell assemblies in a recurrent network. Our results suggest that the brain's ability for few-shot learning is supported, at least partly, by highly sophisticated information processing by single biological neurons.

## Introduction

Evidence indicates that cortical neurons can detect co-activation patterns [1,2] and temporal sequences of presynaptic neurons [3] in a stream of synaptic input. Since spike patterns contain task-relevant information such as sensory stimuli [4] and behavioral contexts [5], computational models have been proposed to understand the mechanism of pattern detection in neurons [6–12] and the roles of such patterns in brain computing [9,13,14]. Experimental studies have addressed the importance of dendritic spikes in coincidence detection [15–17], which is suggested to underlie sequence detection [18] but the mechanisms by which cortical neurons can rapidly become selective to specific patterns of synaptic inputs, especially in the presence of biological noise, remain elusive.

An important distinction across modeling approaches in sequence detection is whether learning occurs in a supervised or unsupervised manner. In supervised learning, a target or supervisory signal is useful to minimize the mismatch between the model response and its expected output. However, evidence points out that the brain can quickly detect salient, though unfamiliar, sensory stimuli [19], suggesting that pattern responses are likely learned in an unsupervised manner. Clarifying whether and how cortical neurons achieve rapid unsupervised pattern learning is important, even if applicable supervised learning rules exist for detailed pyramidal neuron modeling [6].

At least, two different computational approaches to pattern detection in neurons exist in the literature. In one approach, synaptic learning rules derived from experimental evidence such as spike-timing-dependent plasticity (STDP) are investigated for this purpose [4,20–24]. For instance, using experimentally tuned parameters for STDP in an unsupervised neuron model can show how these models become near-optimal in detecting coincidences [9,13,25]. Though such an approach can be informative, obtained results should be interpreted with care since the details of learning rules on which a particular model is based may depend on experimental conditions (culture compositions, single spikes or bursts, dendritic recording sites, etc.). An alternative approach is more goal-oriented, for instance, attempting to implement machine learning-based algorithms in simplified neuron models. These models typically achieve high performance across tasks because its learning rules are

explicitly designed to optimize predefined objective functions, frequently leveraging large datasets and supervised feedback signals. Neuron models that incorporate dendritic mechanisms for learning are of particular interest as they can perform pattern detection and sequence learning across a variety of tasks [8,11].

However, many of these biologically or mathematically well-founded models require an unrealistic number of pattern presentations for learning. In addition, the mathematical models are computationally costly, rely on complex mathematical derivations, learn under non-biological assumptions, or fail to provide explanations of how they converge to their target output, among other weaknesses. Hence, how biological neurons rapidly detect and integrate groups of synaptic inputs that repeatedly depolarize their dendritic tree remains elusive. In this study, we present a novel synaptic plasticity rule that explains how neurons can rapidly adapt their spiking responses to temporal patterns of input in a self-supervised manner under biological assumptions and constraints. Our proposed biological rule was initially inspired on a machine-learning-based rule derived from the hypothesis that the dendritic synaptic activity learns to predict the somatic spike response [8,11,12,26]. The rule demonstrated excellent performance in pattern detection, but it required an unrealistic number of presentations, and the proposed view also lacks strong neurobiological support.

To overcome these issues, especially the difficulty of slow learning, we propose a Hebbian-like synaptic plasticity rule and test it in two-compartmental neuron models that incorporate several biophysical mechanisms suggested to facilitate learning. We further examine the rapid learning capability of our plasticity rule in a recurrent network model embedding pre-configured cell assemblies. Evidence from sensory [27], motor [28], and memory processing [29,30] suggests the active role of such assemblies of cortical neurons in learning novel experiences. We show that these cell assemblies give a reservoir of activity patterns available for encoding novel experiences and dramatically speed up the detection and learning of temporal communities of presynaptic neurons. Surprisingly, just a few repetitions of spike patterns are sufficient for robust learning.

## Results

Previous neuron models based on machine learning (ML) algorithms suggested that learning in single cortical neurons is highly sensitive to statistically salient input patterns [11,12,26]. Here, a neuron model implementing unsupervised minimization between somatic and dendritic activities is particularly interesting as it demonstrated self-supervised chunking of temporally structured spike inputs to perform various cognitive tasks [8]. Below, we first introduce a description of stereotypical salient input pattern generation and then move on to how to implement a biologically plausible synaptic plasticity rule to learn such patterned input. We look into the mechanisms underlying such ML-based learning in a two-compartment neuron model, and use them to clarify the conditions enabling the rapid detection of activity patterns (within several exposures or presentations), which was not the case in the ML-based model.

### Temporal input pattern description

To process sensory information, the brain employs diverse encoding strategies. Neural coding refers to the representation of information in spike patterns, where the spiking activity of projecting neurons and their relative timing carry the encoded information. Low-level sensory features—such as frequency, intensity, and latency—are relatively straightforward to encode by first-order sensory neurons, which emit spikes with probabilities proportional to the encoded property. For instance, retinal ganglion cells encode visual information based on the very first spikes [31], while trigeminal ganglion neurons in the whisker sensation pathway encode texture proportional to the degree of coarseness [32]. In contrast, higher level features projected from thalamic structures to cortical areas cannot be captured by spiking frequency or timing alone. Encoding of complex features, such as edges, shapes, and color in vision or duration, onset, and spectral content in audition, emerges from the intricate interplay of neuronal connectivity and the precise activation of specific neural circuits [33], building on lower-level sensory attributes [34].

A key characteristic of cortical spiking is its high irregularity. To model stereotypical temporal patterns of presynaptic activity, these units can be represented as homogeneous Poisson processes [35], characterized by their firing rate $r$. Spikes are generated by drawing uniformly distributed random numbers at each time step and evaluating the condition $rand() \leq r\delta t$, where $\delta t$, the simulation time step, is sufficiently small such that $r\delta t \ll 1$. With this in mind, we can consider biological aspects such as refractory period or bursting (firing of many spikes in succession) by including these features in the generation process. Bursting of afferent inputs, when included in this work, is only present in patterns.

## Spike trace-based model for self-supervised pattern detection

Supervised [11] and unsupervised [8] learning rules implemented in single neurons hypothesize that the dendritic activity driven by synaptic inputs attempts to predict the somatic response in a statistical sense. Synaptic weights $w$ on the dendrites undergo plastic changes to minimize errors (or information loss) between somatic and dendritic compartment activities. We approximately implement self-supervised learning for pattern detection in a two-compartment model subject to a Hebbian-like learning rule (Fig 1A), which modifies the weight of synapse $i$ by using a "self-supervising" signal $PI(i,t)$ given to the dendrite. The somatic compartment is a leaky integrate-and-fire unit, that is, whenever the somatic compartment membrane potential reaches a threshold, a spike is artificially produced and the neuron resets and enters a refractory state. The dendritic compartment receives the afferent input and its also leaky with coupling conductances connecting both compartments for each direction (dendrite-to-soma and soma-to-dendrite). Separating the neuron model into two compartments is essential in models like ours: the dendritic compartment does not produce spikes but can depolarize to much higher levels than the somatic compartment, offering a broad dynamic range for input integration.

We describe the core of our biological learning rule below. The self-supervising signal has three components. 1) A short-term excitability signal $e(t)$ that entirely depends on the postsynaptic somatic response, 2) postsynaptic potentials (PSPs), which are generated in response to presynaptic spikes (e.g., see Fig 1B), and 3) a weight scaling function $\zeta(|w_i|)$ shown in Fig 1C. The following Eq 1 to 3 and Eqs 16-19 (Methods) describe the proposed learning rule:

$$
\begin{aligned}
PI(i,t) &= e(t)\text{PSP(i,t)}\zeta(|w(i)|), \\
e(t) &= Y(t) - \overline{Y(t)},
\end{aligned}
\tag{1}
$$

where $Y(t)$ represents a graded spike count (decaying with a time constant $\tau_Y$) of somatic spike train $S(t) = \sum_{\text{spikes}} \delta(t - t_{\text{spikes}})$ and $\overline{Y(t)}$ is the low-pass filtered spike trace obtained by solving Eq 2 and Eq 3. These spike traces are only utilized for the plasticity rule and their instantaneous values do not affect the membrane potential dynamics.

$$
\frac{dY(t)}{dt} = \frac{S(t) - Y(t)}{\tau_Y},
\tag{2}
$$

$$
\frac{d\overline{Y(t)}}{dt} = \frac{Y(t) - \overline{Y(t)}}{\tau_{\overline{Y}}}
\tag{3}
$$

The synaptic scaling term $\zeta(|w(i)|)$ is derived from an Alpha function with a unitary range (Eq 17 in Methods) and is designed to dynamically constrain synaptic strength during learning. This function shapes the synaptic weight distribution to match physiological long-tailed distribution of spine head volumes observed under spontaneous activity in hippocampal CA1 neurons [36]. $\zeta(|w(i)|)$ does not directly govern pattern-tuning capacity, it ensures that small synaptic weights have a greater probability of potentiation, whereas large weights saturate and are unlikely to increase further. This mechanism captures the volatile and rapid dynamics of physiological spine growth [37], making spine volume a suitable proxy for regulating synaptic strength.

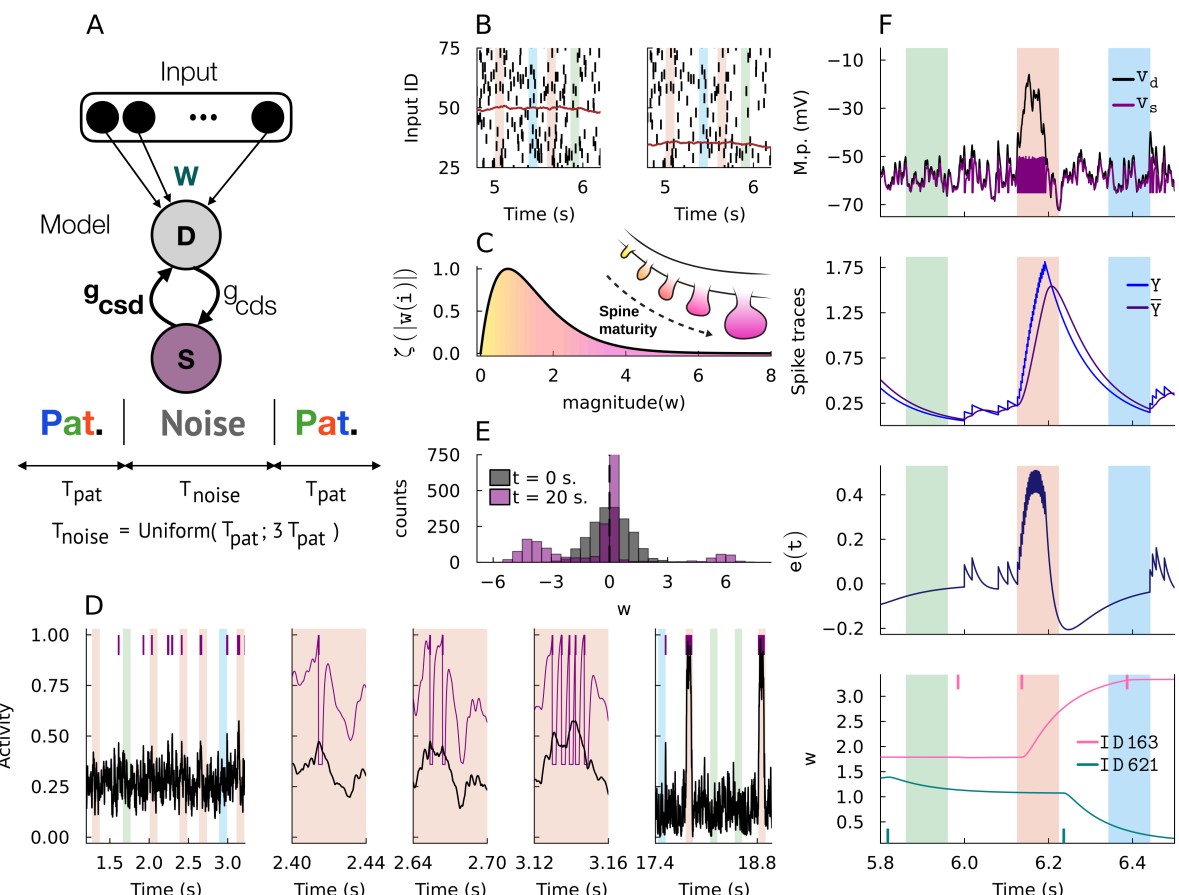

**Fig 1**. **Spike-trace-based synaptic learning rule.** A: The two-compartment neuron model and input scheme. S: somatic compartment. D: dendritic compartment. Input: Poisson spiking units. w: synaptic strengths vector. $g_{cds}$: dendrite-to-soma coupling conductance. $g_{csd}$: soma-to-dendrite coupling conductance. Patterned spike inputs of length $T_{pat}$ occur randomly in temporal input spike trains and are separated by noise spike patterns of length $T_{noise}$ sampled from a uniform distribution ranging from $T_{pat}$ to 3 $T_{pat}$. B: Examples of sent (left) and received (right) spike trains are shown. Due to transmission failure, these spikes trains have different mean spike counts (brown colored traces, same valued y-axis). C: A $\zeta$-scaling function was fitted phenomenologically based on spine maturity distributions. D: Normalized dendritic activity (black) and somatic activity (purple spikes and traces) at different moments during a trial simulation. Note that somatic activity resets when it reaches threshold. E: Synaptic weight distributions at start (t=0) and end (t=20) of a trial simulation. F: Time courses of various dynamical variables of the spike-trace-based plasticity rule are shown. From top to bottom: the somatic and dendritic membrane potentials, spike trace $Y(t)$ and its averaged (low-pass filtered) trace $\overline{Y(t)}$, short-term excitability signal $e(t)$, and the weights $w$ of two example synapses. Vertical colored bars indicate the periods of patterned inputs. Input spikes occurring during positive and negative values of $e(t)$ are potentiated or depressed, respectively.

Finally, before inducing long-term changes in synaptic weight $i$, we low-pass filter $PI(i,t)$ with time decay constant $\tau_\Delta$ and update the synaptic values at a learning rate $\eta$ (Eq 18 in Methods). S1 Table and S2 Table summarize the values of the parameters for synapse and neuron models.

Unless otherwise stated, throughout this study, we test the performance of our model in a temporal pattern detection task depicted in Fig 1A. An input layer consists of 2000 neurons (approximately 1000 excitatory and 1000 inhibitory units), all of which generate homogenous Poisson spike trains as described in the previous section with $r = 5$ Hz. The trial lasts 20 seconds and involves repeated spike patterns of multiple communities of neurons (color-shaded domains in Fig 1B). Each input community consists of about 500 excitatory and inhibitory neurons and is mutually non-overlapping in time. S1A Fig showcases example patterns generated in this way as well as with bursting units. In this study, we use three

distinct patterns (created in the same way), although increasing the number of patterns does not change the essential results. Additionally, synaptic delays were omitted in all our simulations, and all synaptic effects were modeled as instantaneous relative to the simulation timestep; this simplification does not affect the tuning results, which depend on dynamics occurring over slower timescales. Consecutive repeated patterns are separated by non-repeated random spike trains generated with the same rate $r$ (non-shaded domains in Fig 1B) of which duration is sampled uniformly, as shown in Fig 1A. The repeated patterns have an identical duration $T_{pat}$ of 100 ms and occur with equal probabilities.

Our learning rule rapidly increases the detection capacity of repeated structured input, as demonstrated in Fig 1D showing changes in the output spike count of the model neuron before, during, and after an example trial. The model neuron eventually learned a selective response to one of the repeated patterns. There was no explicit target pattern for the neuron model, but the nature of the learning rule allowed it to pick up any of the statistically salient patterns hidden in the input stream. Surprisingly, only several times of exposure to the pattern sufficed for this learning. We regarded that convergence was reached when the model was capable of responding robustly to said pattern. Since pattern templates were created randomly, the neuron model developed either an early stimulus onset (0 to 50 ms) or late stimulus onset (50 to 100 ms) response within the pattern time $T_{pat}$ and, in both cases, satisfied our pattern detection criteria.

We also compute a pattern selectivity index (see Methods) and obtain this metric across time, based on the changes in synaptic weights from pattern weights and non-pattern weights. In S1B Fig, we exemplify how this metric develops. Here, the pattern-colored traces showcase how the green pattern tuning response gradually develops at each pattern presentation. Pattern tuning is sensitive to the initial conditions of the synaptic weights. To see this, we compared synaptic input biases between detected and non-detected patterns in different initial synaptic weight distributions (S1C Fig; see Methods for further details). We found that, despite the randomness of the all-to-all feedforward input structure, patterns that were later detected tend to exhibit significantly higher initial synaptic weights compared to non-detected ones (Mann–Whitney U test, all $p<10^{-4}$). This effect is strongest for the LogNormal and Uniform distributions, where the bounded and skewed nature of the weights exacerbates the impact of early differences. Gaussian distributed weights, though visually overlapping, also displays a statistically significant difference, indicating that even symmetric initialization can introduce measurable bias. Additionally, two further example simulations are presented in S1D and S1E Figs with the latter case having afferent input with 10% percent of bursting units.

When learning converged, the weight distributions of excitatory and inhibitory synapses obtained at the end of the simulation displayed bimodality (Fig 1E). The strong selectivity of the learned neuronal response is hinted from strong excitatory synapses mediating the detected input pattern and strong inhibitory synapses mediating non-detected input patterns. When combined, these distributions yield a long-tailed distribution with dense weak links and sparse strong links. Cortical synapses are known to obey long-tailed strength distributions [38] and their computational implications have been studied extensively in memory encoding and retrieval [39]. In our model, the growth of synapses is controlled by the scaling function that is multiplicative with the Hebbian term (see Fig 1C and Eq 17 in Methods). Pattern selectivity is further hinted in synaptic distributions of pattern synapses across different time snapshots during training (S2 Fig).

## Relationship to machine learning-based learning rules

In the original ML approach [11], synaptic plasticity is governed by an instantaneous rate prediction error, $S(t) - \phi(V_w^*(t))$, which is given by the somatic spike train $S(t)$ (driven externally by a teaching signal) and dendritic prediction of the actual somatic firing, $\phi(V_w^*(t))$, calculated from the rescaled dendritic membrane potential $V_w^*(t)$ through a non-linear filter $\phi$. Similarly, we can derive an unsupervised learning model from the supervised one by removing the teaching signal and introducing an additional mechanism to regulate the cell's intrinsic excitability [8]. By construction, these models are successful in learning non-random, hence salient, statistical features (e.g., repeated spike patterns) hidden in synaptic input. However, they are built on mathematical assumptions of which biological reality is unclear and require a large number of input

presentations to converge. Several questions remain to be answered: can single neurons calculate the probability distributions of somatic and dendritic activities and the distance between them, which underlie the predictive learning rule? What biological mechanisms are required for this? More specifically, the previous models assume that the dendrite has a readily available signal that predicts somatic output. Such a signal is particularly crucial for unsupervised learning as it solely relies on it. Finally, are thus-constructed biological neuron models equally (or even more) efficient in detecting and learning non-random input features?

The learning rule proposed in this study aims to address these questions. Moreover, while sharing certain similarities, the present plasticity rule can solve the difficulty encountered by the ML-based unsupervised learning rule. Unlike the ML-based rule that relied on a memory trace of presynaptic inputs, our short-term excitability signal $e(t)$ depends on a memory trace $\overline{Y(t)}$ of recent postsynaptic activity. The dependency on the history of synaptic inputs or that of postsynaptic firing forces the neuron's current spiking activity to obey the probabilistic structure of learned salient synaptic inputs. Thus, these memory traces implicitly represent the predictive nature of the learning rules. However, contrary to prior work, where models learn by explicitly predicting future somatic events and error signals decrease across presentations, our proposed signal *increases* when repeatedly activated presynaptic patterns forces somatic spiking. These synapses can increase until they are constrained by the synaptic scaling function $\zeta(|w(i)|)$.

One issue in the ML-based unsupervised rule [8] is the trivial solution of all weights converging to zero ($w \to 0$), because this gives the easiest case where no spiking output is predicted for no synaptic input. In contrast, our biological rule can avoid this difficulty by producing sustained fluctuations around a baseline output activity. The short-term excitability signal $e(t)$ serves as a proxy for recent variations of postsynaptic activity: at any given time when $e(t)$ is positive, all inputs that contributed to spiking are potentiated by a sampled fraction of this signal. Positive deviations thus suggest recent increases in neural activity, reflecting bursts or responses to stimuli. When $e(t)$ becomes negative (typically when spiking has recently ceased), any input after it will be depressed. Such negative deviations then reflect recent decreases in neural activity due to inhibition or moments of quiescence. During moments of silence, $e(t)$ relaxes back to zero, yielding no synaptic weight changes. The total amount of synaptic plastic changes is governed by the temporal dynamics of the plasticity signal $e(t)$. While LTP and LTD do not necessarily balance pointwise, the temporal average of $e(t)$ over the stimulation window approximately vanishes when integrated from stimulus onset to a sufficiently long post-stimulus duration, due to its relaxation to zero. This ensures that, on average, potentiation and depression contributions equilibrate over time.

**Facilitation of temporal community detection by transient boosting of somato-dendritic coupling**

The results shown in Fig 1 demonstrated that the proposed learning rule enables the two-compartmental neuron to detect temporal activity patterns rapidly. To accelerate the pattern detection, this learning rule uses a plasticity signal that is updated instantaneously for every somatic spike and becomes available at all active synapses, as in STDP, which gives a convenient mechanism of pattern detection [9,13,25,40]. However, such a signal is not readily available in biological cells. Therefore, we now construct a slightly more plausible model that takes into account the mechanistic function of backpropagating action potentials (bAPs) [41,42].

In the following model, we introduce high-voltage activated (HVA) $Ca^{2+}$ channel dynamics [43] in the dendritic compartment (see Methods for channel equations) and an intracellular $Ca^{2+}$ concentration trace via integration of calcium current $I_{Ca}$ (Eq 4). It is known that postsynaptic $Ca^{2+}$ concentration dynamics are a proxy of somatic activity, especially in the basal and in tuft dendrites [44], and have a pivotal role in synaptic plasticity [45].

$$\frac{dC(t)}{dt} = \phi I_{Ca} - \left( \frac{C(t) - C_0}{\tau_C} \right). \tag{4}$$

The parameter $\phi$ scales $I_{Ca}$ into $Ca^{2+}$ concentration, $C_0$ is the $Ca^{2+}$ concentration at rest, and $\tau_C$ is the time constant of $Ca^{2+}$. We swap this $C(t)$ trace for the spike train trace $Y(t)$ in the synaptic plasticity rule. Similarly as in the previous rule, calculating an averaged trace $\overline{C(t)}$ by low-pass filtering $Ca^{2+}$ concentration, we modified our proposed signal for the plasticity rule shown in Eq 5:

$$
\begin{aligned}
PI(i, t) &= e_c(t)\text{PSP(i,t)}\zeta(|w(i)|), \\
e_c(t) &= C(t) - \overline{C(t)},
\end{aligned}
\tag{5}
$$

The rest of the equations (Eq 17-19 in Methods) in the plasticity rule remain unchanged.

Plugging in HVA-$Ca^{2+}$ channel dynamics into the dendritic compartment is relatively straightforward and its effect on dendritic depolarization is minimal (see S1 Table for parameter values). With this change, synaptic plasticity updates can now occur, though small, even in the absence of postsynaptic spiking. However, the calcium-based excitability signal $e_c(t)$ is not directly linked to somatic spiking activity and fails to reach a target pattern. We showcase an example failure trial where, compared to the spike-based model (Eq 1), the calcium-based plasticity alone is unable to converge (Fig 2A). Thus, the calcium-based plasticity mechanism alone is less efficient than the naive mechanism having direct access to the somatic spiking.

We introduce a transient boosting in somato-dendritic coupling to rescue the $Ca^{2+}$-based model and recuperate its learning capacity (Fig 2B). During a backpropagating event in biological cells, a wave of depolarizing current travels across the dendritic tree. We mimic this biophysical phenomenon by transiently increasing the coupling conductance from somatic to dendritic compartments (i.e., $g_{csd}$ in Methods Eqs 29–30). This key behavior only occurs during somatic spiking and is abrupt enough to avoid noisy somatic currents in the dendrite from sub-threshold somatic fluctuations.

In this sense, a large constant somatodendritic coupling would disrupt the system's stability and specificity. It would either cause the coupled system to blindly follow fluctuations, where dendritic variability leads to somatic spiking and further propagates back to the dendrites without regard for input patterns, or, in cases of bursting activity, the excessive coupling would amplify the feedback loop to such a degree that the system becomes unstable and unable to terminate its own activity. This runaway effect undermines the precise temporal control required for learning. Conversely, a coupling value that is too small would fail to sufficiently amplify relevant signals, preventing the induction of meaningful synaptic plasticity. By using a transient boosting mechanism, the coupling is dynamically increased to align with functionally significant spiking events, ensuring both stability and pattern-specific learning.

As shown in Fig 2B, the transient boosting in $g_{csd}$ dramatically improves both speed and signal-to-noise ratio (SNR) of pattern detection by the $Ca^{2+}$-based model. The result of the transient increase is an intrinsic positive feedback formed by the somato-dendritic coupling and dendritic $Ca^{2+}$ dynamics. With somatic spiking, a transiently boosted somato-dendritic coupling further depolarizes the dendritic membrane potential. This causes an influx of calcium current through the HVA-$Ca^{2+}$ channel, serving an increase of $Ca^{2+}$ concentration that raises the magnitude of plasticity updates. We note that the upswing dendritic membrane potential makes calcium-based plasticity traces resemble $Y(t)$ and $\overline{Y(t)}$.

As far as the somato-dendritic coupling generates sufficiently strong bAPs, our proposed rule with spike-based signals shows excellent learning performance. We performed 500 trial simulations and assessed the model' performance by calculating the SNR at different bAP strength (measured by $m_{csd}$ given in Methods Eq 30). We measured the signal (S) as the number of spikes occurring during the tuned pattern presentation and noise (N) as otherwise, i.e. any other spike outside of the tuned pattern. The results presented in Fig 2C indicate that the higher the bAP pulse, the stronger the performance of the model and its capacity for detection. However, the induced dendritic depolarization can grow out of proportion for extreme values of bAP pulse, increasing the number of outliers in the low SNR regime and, sometimes, even surpassing the dendro-somatic conductance $g_{cds}$ (Fig 2D). We note our approach assumes a spike-triggered transient increase for $g_{csd}$ but no changes in $g_{cds}$ with the exception of refractory period when $g_{cds} = 0$ (see Methods Eq 21).

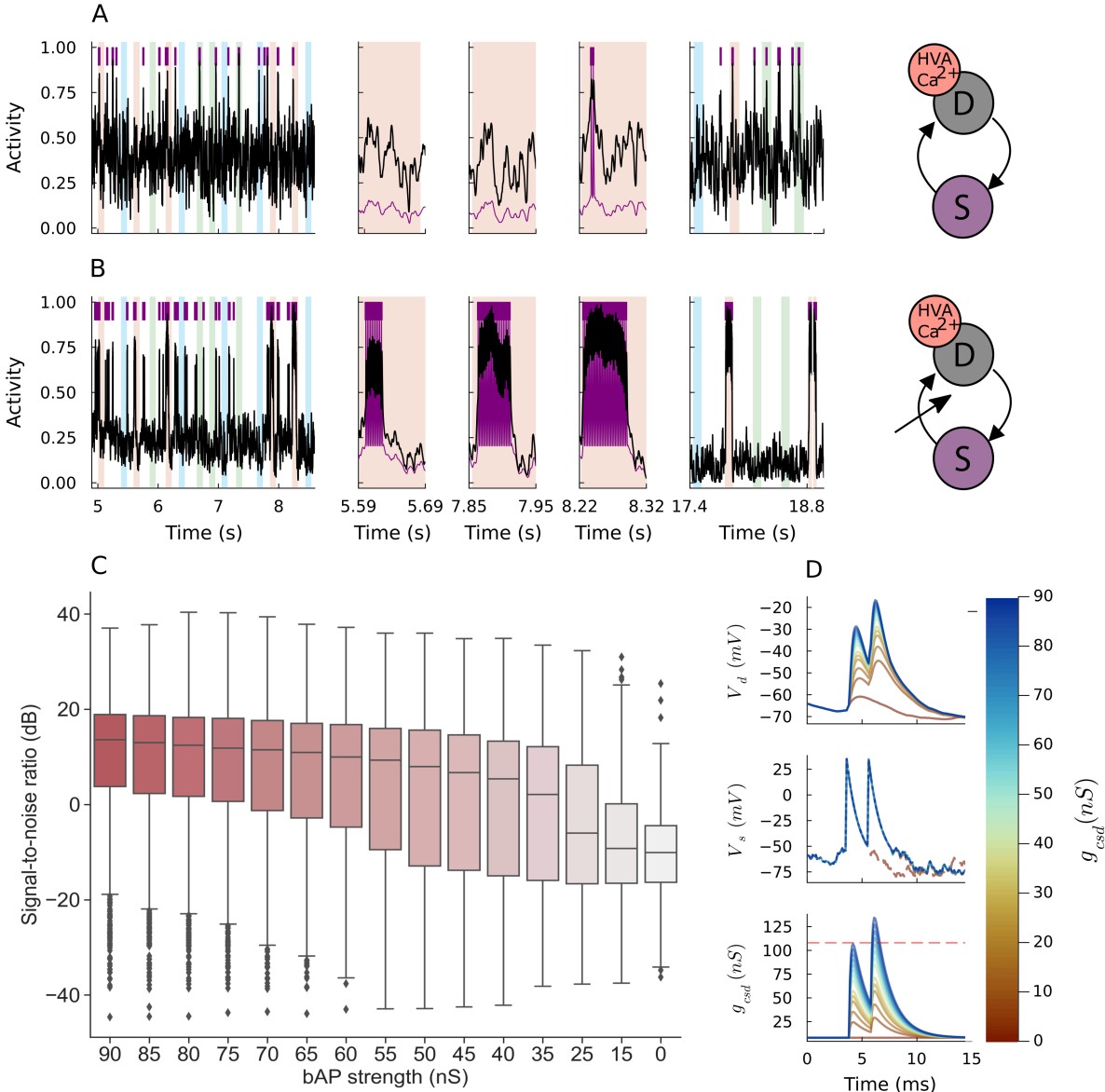

**Fig 2. Calcium-based plasticity rule with spike-triggered transient somato-dendritic coupling boosting.** A: Normalized dendritic activity (black) and somatic activity (purple spikes and traces) at different moments during a trial simulation are shown for the calcium-based neuron model without a transient boosting of the somato-dendritic coupling. B: Same as above but the neuron model includes transient boosting of somato-dendritic coupling during spike generation (the bAP strength of 60 nS). A cross arrow schematically illustrates the transient boosting of the somato-dendrite coupling. C: Box plots containing five-number summary and outliers of signal-to-noise results (500 trials for each case; failure trials discarded) are shown for the Ca-based neuron model with the transient coupling boosting at different bAP strengths. D: Effect of different bAP strength on actual AP backpropagation is assessed. Colors indicate each bAP strength and dashed red line is the dendrite-to-soma coupling. At bAP strength of 15 nS, the dendritic depolarization triggers a second somatic spike.

This implies that there is an optimal range of the bAP pulse value for robust performance. If this value is too small, pattern detection becomes difficult, as demonstrated in Fig 2A for an extreme case of vanishing somato-dendritic coupling. However, if the value is too large, even somatic spike responses to non-target patterns propagate to the dendrite and drive learning. Our model suggests that, in the biological scenario, a realistic bAP event is determined to solve this trade-off.

Excitatory and inhibitory conductances behave in a way that facilitates selective responses to a given pattern. In the example cases presented in Figs 1 and 2 (bAP strength 60 nS), excitatory conductance remains high relative to inhibitory conductance during the presentation of pattern red (S3A and S3C Figs), whereas inhibitory conductance overwhelms the excitatory one during the presentation of the other patterns (S3B and S3D Figs), resulting in strong red pattern selectivity. The transient somato-dendritic coupling increase forces the dendritic membrane potential upswings given somatic spiking, which introduces further somatic depolarization post refractoriness.

The choice of including the HVA-Ca$^{2+}$ channel in our model is based on its activation threshold dynamics. There are, however, many other calcium channels that lead to further influx at different depolarization levels and may be useful to incorporate. However, our approach here only regards necessary mechanisms for producing rapid learning of hidden patterns amidst noise. It is an intriguing open question whether including other calcium channels can produce a similar effect to the transient somato-dendritic coupling increase on the pattern detection of the model.

## Crucial roles of NMDARs for rapid binding of patterns with low SNR

One crucial component that has a strong influence on the long-term potentiation and long-term depression of coincidently active synapses is the activation of NMDARs in dendritic spines [46]. An increased number of NMDA binding sites suggests a strengthened synapse, whereas a decrease number of binding sites is a sign of a weakened synapse [47]. The role of NMDARs in discriminating spatiotemporal activity patterns was also reported experimentally [48]. We investigated what is the effect of including NMDAR channel dynamics in the dendritic compartment using Jahr & Stevens (1990) [49] voltage-dependence, and rise and decay time constants 3.3 and 102.38 ms respectively. The time constant values are temperature-(Q10-)corrected for cortical pyramidal neurons [50], however there is no strict requirement for exact time constant values. The set of equations for NMDAR current has been widely used and introduces sustained dendritic membrane potentials due to their slow timescale. As not all synapses tend to include AMPA, we chose a ratio of NMDA-to-AMPA of around 75% (excitatory IDs picked randomly), which is typically found in Schaffer Collateral synapses in CA1 [51,52]. Additionally, NMDARs are also permeable to calcium, and we model this by attributing 5% of the total NMDA current to the calcium current $I_{ca}$ (see Methods Eq 31). Based on experimental observations at typical external calcium concentrations of 1~2mM [53], this percentage is typically assumed between 5-10% [54]. The NMDAR-mediated calcium current combines with HVA-Ca$^{2+}$ current and is integrated into $C(t)$, thus affecting $e_c(t)$.

In Fig 3A we showcase an example trial for our Ca$^{2+}$/NMDAR-based model. The model now includes HVA-Ca$^{2+}$ channel equations, NMDAR equations, and the variant learning rule with calcium-based plasticity traces (Eq 5) and with the spike-triggered transient boosting of $g_{csd}$. As expected, the neuron model rapidly converges to robust spiking to pattern red. For this case, we observe that the inclusion of NMDA receptor dynamics in the dendritic compartment can result in steeper rising and falling edges of the excitability signal $e_c(t)$, which now captures many more synapses during post-synaptic spiking. However, a question arises: how is the inclusion of NMDARs affecting the neuron's pattern detection performance?

To answer this question, we calculated SNRs for each of the three different models, i.e., spike-trace-based, Ca$^{2+}$-based, and Ca$^{2+}$/NMDAR-based models, where the SNR is defined as in Fig 2C and the latter two models employed the transient somato-dendritic coupling boosting. We calculated SNRs for a range of the bAP strength. For all models, the same learning rate ($\eta$ in Methods Eq 18) was utilized. We plot the average SNR in Fig 3B. The spike-trace-based plasticity rule yields the highest SNR among the three models. More importantly, the SNR of the calcium-based model with NMDAR dynamics (Ca$^{2+}$/NMDAR-based) is nearly matched with the highest SNR. Hence, the latter is the most capable of rapidly detecting patterns whilst remaining biologically plausible. In general, as shown in Fig 3C, all models exhibit similar distributions of excitatory and inhibitory synaptic weights at the end of a given trial simulation ($m_{csd} = 60$ nS trial) where convergence was achieved. However, in the distribution of the Ca$^{2+}$/NMDAR-based model, the peaks of strong excitatory

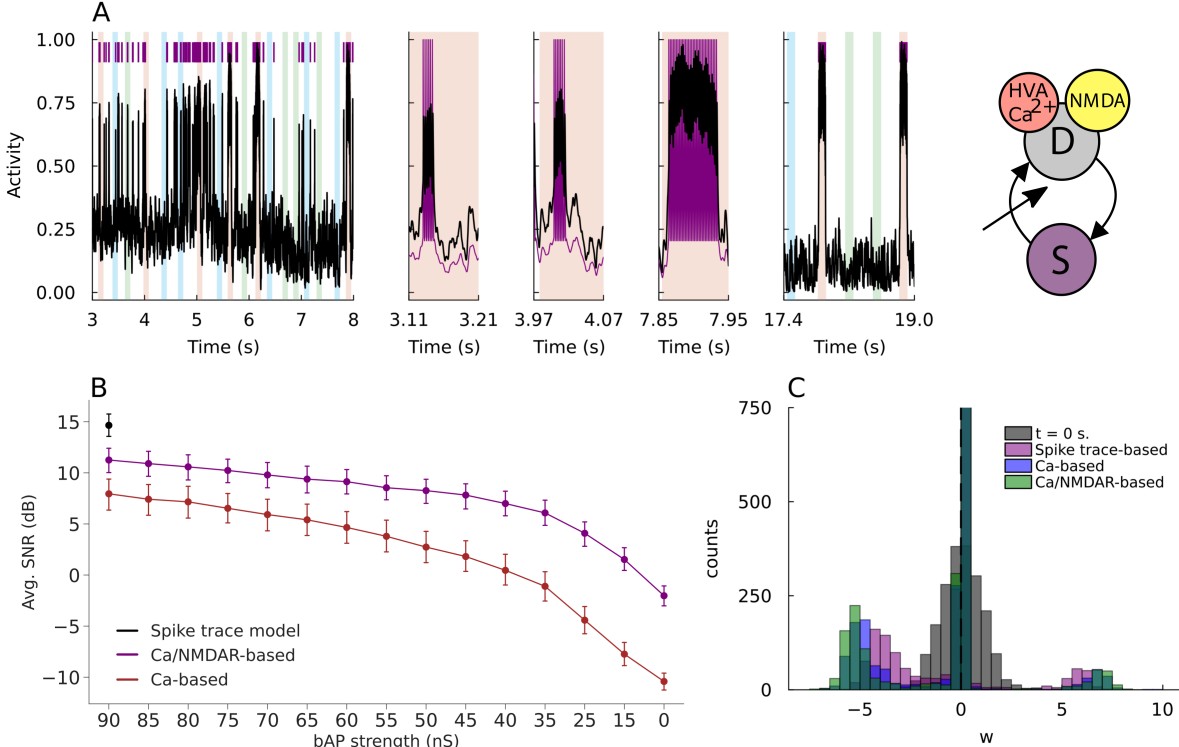

**Fig 3**. **Enhanced detection capability of $Ca^{2+}$-based plasticity rules with NMDA receptors.** A: Normalized dendritic (black) and somatic (purple) activity and spikes at different moments during trial simulation for calcium-based model including NMDAR dynamics. Panels from left to right depict early trial time, three red pattern presentations, and late trial time. Illustration for Ca-based neuron model including NMDA receptor dynamics and spike-triggered transient boosting of the somato-dendritic coupling. B: Point plot of average SNRs of all models resulting from 500 simulation trials (no tuning trials discarded) at different bAP strengths. Error bars indicate confidence intervals (95%). Results of the spike trace-based model are plotted for comparison. Across all values of the bAP strength, the average SNR is significantly higher for the model including NMDAR dynamics. C: Synaptic weight distributions at the start and end of trial simulations for all three models with the bAP strength of 60 [nS].

and inhibitory synaptic weights are most clearly distinct from those of weak synapses, consistently with the excellent performance of this model. Put differently, although models utilizing the calcium-based excitability trace $e_c(t)$ achieve a lower average SNR compared to the artificial spike-based trace $e(t)$, the $Ca^{2+}$/NMDAR-based model demonstrates the highest selectivity by reinforcing a greater number of synapses.

## Robustness of pattern tuning against temporal structure corruption

The learning efficiency of the model depends on various factors, including the number of coincident spikes within each pattern, the initial synaptic weight values, and noise intensity in input patterns. We explored how the model tunes to different patterns in the presence of spike timing jitters, varying the noise intensity. Input spike trains involved three different patterns (named blue, red, and green) with equal occurrence probabilities. Given enough simulation time, the model tuned to a specific pattern almost with equal probability (Fig 4A). For many of these trials, a small number of presentations of the pattern were sufficient to reach convergence (notice graphs on shorter simulation times). In a small fraction of trials, the neuron model tuned to more than one input pattern, and the fraction of trials remained constant as the simulation time was prolonged ("Others" in Fig 4A). This mixed tuning was presumably due to accidental overlaps among input patterns or unfavorable starting conditions.

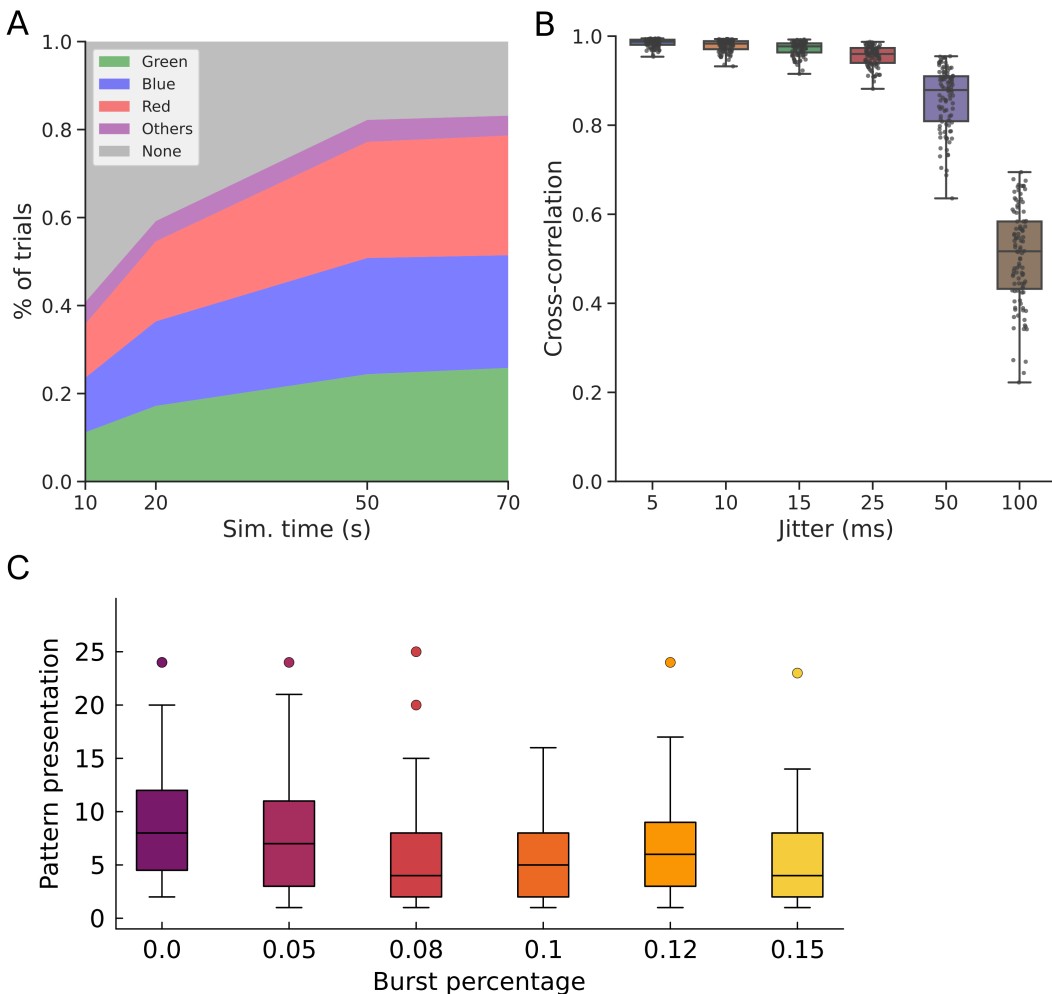

**Fig 4**. **Convergence, robustness, and tuning time of pattern learning task.** A: The average percentages of trials in which the model tuned to particular patterns are plotted for trial simulation lengths of 10, 20, 50, and 70 seconds. Five hundreds (n=500) trials were performed in each case. "Others" label trials wherein more than one pattern was learned by the neuron. B: Robustness test of learned response against temporal structure corruption of learned input pattern via uniform jittering. Box plot overlayed with dot plot of cross-correlations between somatic spiking outputs given uniformly jittered input and the ground truth output are shown for different mean jitters. The data points on each box (n=100 tuned trials) correspond to each 20-second trial where model tuning for a given pattern converged. Box plots show five-number summary measures for each jitter case. C: Convergence assessed by the number of pattern presentations (exposures) vs burst percentage of input units for learned salient pattern (n=100 each) obtained from selectivity index computation once conditions are met (see Methods and S1 Fig).

We highlight the importance of the temporal community structure within input patterns (i.e. temporally structured groups of coincident spikes). While the balance of excitation and inhibition (E/I) varies across each pattern presentation due to synaptic transmission failure ($p_{fail}$), the temporal community belonging to a certain pattern is what governs the learning process. In this sense, synaptic transmission failures were incorporated to emulate experimentally observed synaptic unreliability, introducing trial-to-trial variability that enhances learning robustness by discouraging overfitting to individual spike coincidences. To study how robust the post-learning model performance is, we increased the mean jitter timing of input spikes (Fig 4B). Namely, we selected the results of 100 previous 20-second simulation trials wherein pattern tuning was achieved, and evaluated the robustness of the model's responses against the corruption of the original patterns. We tested the robustness by using a "ground truth" spiking output, which is a response of the post-learning neuron model to a

"clean" input pattern without timing jitters. To prevent synaptic weights from adapting to noisy input patterns, we continued to freeze synaptic weights during the test and repeatedly applied noisy input patterns. We calculated cross-correlations between the ground truth output and outputs to the noisy patterns, revealing that pattern tuning is robustly maintained even for a 50-millisecond mean jitter but drops to a chance level for a 100-millisecond mean jitter, which coincides with pattern duration.

Learning converges relatively fast, requiring only several pattern exposures. To assess the overall tuning performance, we measured how many presentations were required. For this, we obtained the number of presentations given the tuning time at which a pattern selectivity index metric (see Methods and S1 Fig) exceeded the threshold. In Fig 4C, the results show that the neuron model takes on average 7 to 8 pattern presentations during a trial to develop robust output spike bursts when the fraction of bursting units (in %) in salient patterns was varied. We note that the number of presentations required for convergence decreases as the percentage of bursting input units is increased. The decreasing trend is observed until the fraction of bursting units reaches 10%, but not beyond this point. This is presumably because inhibitory units can also exhibit strong bursting activity and block the decreasing trend.

In summary, the model demonstrates robust pattern recognition capabilities up to a threshold of temporal jittering. Notably, the neuron model can converge within 7 to 8 pattern exposures, corresponding to several seconds of simulated time. Increasing the percentage of input spike bursts can accelerate learning convergence.

## Burst-induced Schaffer-collateral STDP replicated by the calcium-based model

STDP protocols reveal important information about the plastic changes in incoming synapses on the postsynaptic neuron. It is widely accepted that protocols involving a single presynaptic spike and a single postsynaptic spike lead to LTP and LTD at excitatory cortical synapses [20,55,56] (see Shouval et al. [23] for review), depending on the relative times of these spikes. However, STDP rules with bursts of spikes have also been studied to reveal different temporal profiles. Then, a question arises: what type of STDP profiles does our proposed learning rule produce? One can conventionally study the learning capabilities of neurons using STDP curves and then model the underlying biological mechanisms later. Contrastingly, we first formulated a biological mechanism for rapid pattern detection in a neuron model. However, STDP curves provide a concise, experimentally interpretable representation of our learning rule's temporal and dynamical properties, serving as a "signature" of how synaptic changes are induced. Therefore, it is crucial to derive STDP-like profiles as emergent properties of our biologically motivated model. By comparing these emergent STDP profiles with experimental data and other models, we can better understand the computational implications and the biological plausibility of our proposed plasticity rule.

It was shown that CA3-to-CA1 excitatory synapses in the hippocampus exhibit Mexican-hat-shaped weight changes when a burst of 2 postsynaptic spikes were repeatedly paired with a presynaptic spike at a frequency of 5 Hz [56]. Namely, LTP and LTD were induced for shorter and longer relative times, respectively. Owing to postsynaptic spike bursts and different experimental preparations (see Discussion for the latter point), this protocol is considered to induce stronger bAPs in CA1 pyramidal cells. This specific STDP protocol is intriguing because it incorporates bursts of postsynaptic spikes that can facilitate effective synaptic plasticity and robust information transfer between pyramidal neurons. Below, we show that our calcium-based rule replicates this STDP rule without assistance from NMDARs. For comparative reasons, we followed the pairing protocols used in the experiment, with a total of 20-30 or 70-100 times of repetitions. We also adopted the convention used in that study, defining $\Delta t$ as the time interval between postsynaptic bursts and presynaptic spikes.

In Fig 5, we show the STDP rules obtained from numerical simulations of the calcium-based model with experimental data points overlaid [56]. To begin with, we examined the STDP rule resultant from 70-100 pairings of single presynaptic and postsynaptic spikes at frequencies of 1 Hz and 5 Hz. We introduced small noise in the model (parameter $\rho$ in S2 Table) to produce basal fluctuations in the error signal $e(t)$, which result in a minimal LTD across all $\Delta t$ values (Fig 5A).

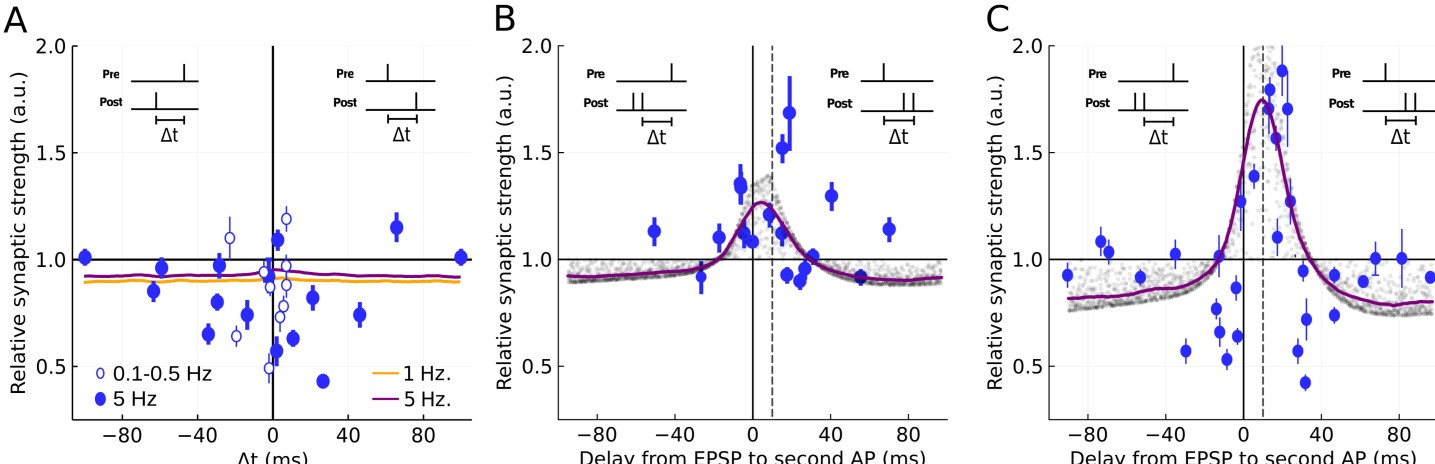

**Fig 5. STDP rules obtained by our Ca$^{2+}$-based plasticity rule.** Each trial has many repetitions at different time delays between a presynaptic spike and a second somatic AP (inset), where positive intervals are defined as causal, and negative intervals as anti-causal. Simulation data points are presented with transparency to indicate time-window shapes. Colored solid lines are smoothed polynomial interpolations of simulation data points by linear least square fitting. Experimental data are shown with error bars indicating confidence intervals for 5 Hz (filled circles) and 0.1-0.5 Hz (open circles) pairing protocols. These data points are included with permission from the authors [56]. A: The STDP rule of the model is shown for 70-100 pairings of singlet pre- and post-synaptic spikes at 5 Hz (purple) and 1 Hz (orange). Simulation data points are not included for visibility. B: The model's STDP rule is shown for 20-30 pairings at 5 Hz of presynaptic spikes and postsynaptic spike doublets. C: The model's STDP rule of the model is shown for 70-100 pairings at 5 Hz of presynaptic spikes and postsynaptic spike doublets.

The experimental results suggest that the relative change in the synaptic strength becomes largest around small time differences ($-40 \lesssim \Delta t \lesssim 40$ in ms). Running said protocol with our proposed rule, the average relative weight change displayed a slight increase around small time differences for both frequencies. However, since the increase was small and experimental data were also scattered, we cannot safely conclude that the results are consistent between experiments and numerical simulations.

In contrast, the simulation of the model neuron for bursts of postsynaptic spikes generates results more consistent with the experiment. When a single presynaptic spike was repetitively paired with a burst-like pair of postsynaptic spikes 20-30 times, pyramidal cells exhibited an LTP-only (or LTP-dominant) time window (Fig 5B). The peak amplitude of LTP occurred near the origin of the time difference, where the spike singlet-doublet pairing protocol shifted the peak position slightly towards the positive (causal) side. Increasing the number of repetitions (70-100 pairings) enhanced the peak LTP amplitude, flanked by LTD sides that returned to baseline fluctuations towards longer time differences (Fig 5C). Testing our proposed rule with this particular pairing protocol produces similar STDP time windows for both low and high numbers of repetitions. These results demonstrate that the proposed model describes a biologically plausible learning mechanism of cortical cells.

In vitro experiments have suggested that the LTD induced by causal pairings of presynaptic and postsynaptic spikes may be attributable to the presence of inhibition [21,22]. In the present simulations, the LTD sides that flank the LTP time window do not fit the experimental data points well. We suspect that the causes of these LTD time windows are related to biological mechanisms that have yet to be included in our model. These mechanisms seem to require a higher number of repetitions irrespective of the number of post-synaptic spikes, whether they are singlets (Fig 5A) or doublets (Fig 5C). The nature of the synaptic rule presented here tells us that on singlet input-output pairings, the somato-dendritic coupling boosting is not enough to activate HVA-Ca$^{2+}$ channels that would increase Ca$^{2+}$ levels when single spikes occur. This implies that the present model lacks plasticity mechanisms for the induction of strong LTD at low stimulation frequencies.

Nevertheless, strong LTD can still occur when the plasticity signals relax after sufficiently many bursts of spikes (S4 Fig), which homeostatically regulate synaptic conductance changes.

### Few-shot pattern detection by pre-existing cell assemblies

Now, we investigate whether pre-existing assemblies of the proposed neuron model improve the detection of temporal communities in afferent inputs. For this purpose, we constructed a recurrent network of 400 excitatory and 100 inhibitory units with 20 pre-configured excitatory cell assemblies with average size of about 18 units each. To reduce the load of numerical simulations, we used the spike trace-based two-compartment model defined in (Eq 1). For a comparative reason, we used the same input scheme as used in Fig 1A, which contained three repeated patterns. Connections from excitatory and inhibitory afferent input units to network units are all-to-all, with Gaussian-distributed synaptic weights (Fig 6A). All excitatory and inhibitory connections in the network model were modifiable according to the proposed learning rule. We evaluated the ability of the recurrent network to detect any of the salient patterns amidst noise. Network parameters used are presented in S3 Table. Network synaptic strength (E→E, E→I, I→E, I→I) parameters were drawn from lognormal distributions with parameters obtained from experimental data [57].

With appropriate network connectivity and initial synaptic weight distributions, the network self-organized robust response patterns in a competitive manner, wherein neurons belonging to a pre-existing assembly were selectively co-activated by one of the salient input patterns. To support stable dynamics at the onset of simulations, we incorporated relatively strong excitatory-to-inhibitory (E→I) connectivity, as typically observed in cortical circuits, to ensure that inhibitory neurons are reliably recruited by excitatory activity. This initial E/I bias prevents runaway excitation and provides a scaffold for balanced activity. As stated earlier, the plasticity rule continuously updates both excitatory and inhibitory synapses, allowing the network to self-organize into a novel, pattern-driven balanced state different from the initial configuration.

In Fig 6B, cell assemblies could learn such selective responses at the early stage of learning as long as their within-assembly connections were strong enough. Due to the nature of the recurrent connectivity, it was rarely the case that a single assembly of neurons grew up to dominate the activity of the entire network. For instance, a highly active assembly of neuron ID 104 to neuron ID 127, which became selective to none of the patterns, neither impeded a strong response to pattern red of the assembly with neuron IDs 62 to 85 nor the response to pattern blue of the assembly with neuron IDs 180 to 201. To access the rapid learning by pre-existing cell assemblies quantitatively, we show the spike count occurring during each pattern presentation across the 10-second simulation for the example trial (Fig 6C, left). Quantitatively, we found that the mean pairwise correlation within assemblies was 0.59, while between-assembly correlations were significantly lower (0.09), supporting the emergence of selective synchrony within tuned groups (S5 Fig). To our surprise, very few presentations were sufficient to develop robust responses to salient patterns. The growth of neuronal responses is also indicated by the responses of cell assemblies with neuron IDs 62-85 and 180-201 during the early and late phases of pattern learning (Fig 6C, right). However, some pre-existing cell assemblies could learn no selective responses to patterns, as exemplified by the assembly with neuron IDs 104-127 in Fig 6B. Because the network could not learn robust responses without the plasticity of inhibitory neurons, inhibitory plasticity is likely to be crucial for the stability of pre-existing cell assemblies and the formation of their pattern-selective responses.

While few-shot learning of patterns is remarkable, it is unclear whether strongly coupled assemblies are necessary for such a fast convergence. To address this, we run a set of 20 networks endowed with the same connectivity in two different conditions (i.e., 40 trial simulations in total). In a scenario, excitatory assemblies are strongly connected (as is the case in Fig 6B) with Gaussian distributed weights with the mean $\mu = 3.0$ and standard deviation $\sigma = 1.0$, while in a second scenario, within-assembly synaptic weights are weak and defined with Gaussian parameters $\mu = 0.0$ and $\sigma = 1.0$ (the weights are always positive). The spike counts normalized by assembly size are plotted against the number of pattern (any) presentations in Fig 6E. Spike counts are normalized by cluster size to control for variability across networks. The size of the assembly remains constant under all conditions and throughout the trials. Overall, the strongly coupled pre-existing cell

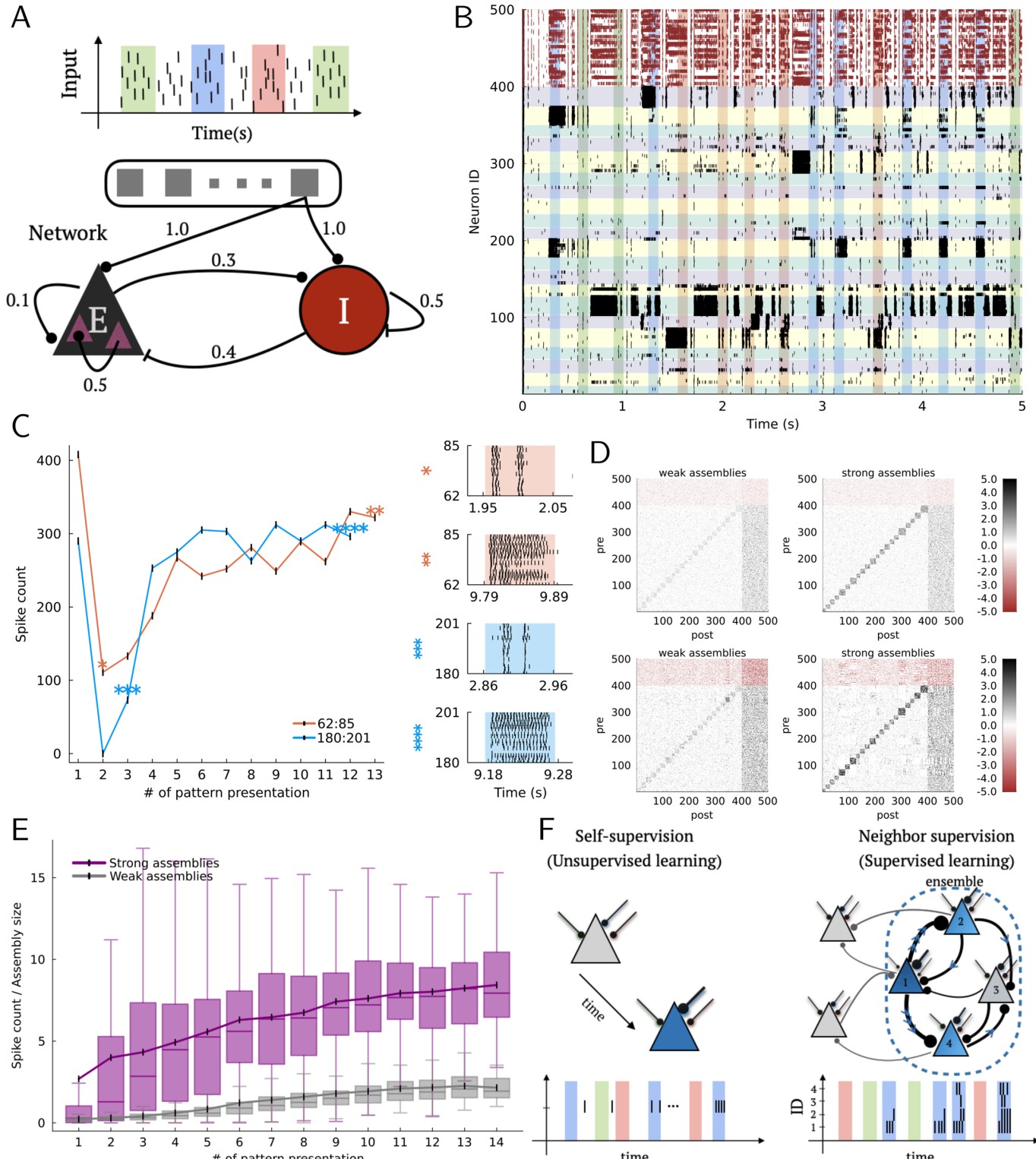

**Fig 6. Few-shot learning through competition among pre-existing assemblies.** A: Recurrent network with pre-existing cell assemblies receive all-to-all afferent input from an input layer used in Fig 1A. Numbers indicate connection probabilities. B: Raster plot of network activity in 10-second trial. Only first 5 seconds are shown. IDs 1-400 are excitatory (black) and 401-500 are inhibitory (red-brown). Presentations of patterns are indicated as vertical

spans and pre-existing assemblies as horizontal spans. C: Left, Spike counts of two cell assemblies across tuned pattern presentations. Asterisks matched to the corresponding right panels. Right, Raster plots of assembly activity for tuned pattern presentation. D: Heatmaps of synaptic weights at start (top) and end (bottom) of trial, truncated between −5 and +5 for visualization. E: Box plots (outliers excluded) and mean value of tuned assemblies' spike counts relative to assembly size for strongly (purple) and weakly (gray) coupled assemblies (20 trials each) across presentations. F: Neighbor supervision paradigm: ensemble dynamics as implicit supervisory signals.

assemblies are capable of producing, on average, a robust spiking response from the very beginning of the simulation whereas the weakly coupled ones require many more presentations. In terms of time-to-convergence, the number of presentations required for weakly coupled assemblies generally matches that of the single-cell model presented in previous figures. The ability of pre-existing cell assemblies to develop a robust spiking response to statistically salient patterns is further enhanced by the strengthening of with-in assembly excitatory connections. As indicated by the heatmaps of the recurrent weight matrix before and after training in each condition (Fig 6D), such strengthening occurs in strongly-coupled pre-existing assemblies but not in weakly-coupled ones. We also observe that cell assemblies become tightly coupled with the recurrent inhibition that enforces network stability.

We propose that within neural networks, certain neurons consistently exhibit heightened readiness to spike, irrespective of their initial excitability baseline or the distribution of afferent connections. Upon receiving structured presynaptic activity, these neurons can swiftly discern statistical significance, facilitating accelerated learning. We term this phenomenon 'neighbor supervision' (Fig 6F), wherein select neurons within an assembly assume the role of supervisors or teachers, augmenting the excitability of their counterparts. This mechanism parallels supervised learning, wherein the target information from the structured presynaptic community is implicitly communicated among assembly members.

In summary, we found that the proposed learning rule implemented at afferent and recurrent synapses enables few-shot self-supervised learning of temporal communities of presynaptic neurons. Pre-existing cell assemblies embedded in a recurrent neuronal network further accelerate this pattern detection. Importantly, the plasticity of inhibitory synapses, which we assumed obey the same proposed learning rule, is necessary for stabilizing network responses to patterned synaptic inputs.

## Discussion

In this study, we have proposed a synaptic plasticity rule for cortical neurons and implemented it in a two-compartment neuron model. The learning rule is inspired by machine-learning-based rule for single cortical neurons and supported by findings in cortical neurobiology. Like a machine-learning counterpart, the two-compartment model is endowed with computational capacity to detect and robustly spike to hidden patterns amidst noisy input. Our modeling work predicts that a transient boosting of the somato-dendritic coupling is sufficient for biologically-plausible pattern detection and NMDARs significantly improve the SNR of this learning approaching the ideal case with the spike trace-based rule. The proposed learning rule can replicate the STDP rule observed for spike bursts at the hippocampal Schaffer collateral, and furthermore, pre-existing cell assemblies greatly accelerate pattern detection in a recurrent network of two-compartmental neurons.

### A biologically plausible single-cell mechanism to learn activity patterns

The capacity of PCs to effectively adapt their neuronal responses to stimuli is largely mysterious. Various plasticity rules have been proposed to understand how PCs assign credit to the relevant synapses. Additive STDP leads to efficient close-to-optimal pattern detection on single neuron modeling [9]. Voltage-based STDP rule supports pattern detection in models with plausible neural circuitry, and dendritic voltage-based STDP preserves learned associations across longer periods of time [10]. STDP is particularly useful in competitive neural networks [13,40]. While STDP rules may be a simplistic abstraction of the complex biological processes that underlie synaptic plasticity events, machine learning-based

approaches and the resultant nonlinear Hebbian rules aim at optimal performance on complex tasks [6,8,11,12]. Our work based on self-supervised learning sits among this class of algorithms while offering a biologically plausible realization, thus bridging the two classes of synaptic plasticity rules without a predictive framework in mind.

Specifically, we highlight that our biological learning rule does not require an extensive presentation of patterns to credit the relevant synapses rapidly. To achieve this rapid learning, our model hypothesized a spike-triggered transient boosting in somato-dendritic coupling, which enables bAPs to propagate into the dendritic compartment. Cortical PCs exhibit a high correlation between somatic and dendritic activities, and bAPs are suggested to underlie associative plasticity [24,58]. Although this transient boosting needs to be confirmed by experiments, the somata and dendrites of *in-vivo* L5 PCs display widespread coupling asymmetry and high correlation of $Ca^{2+}$ transients especially during somatic burst spiking [59–61]. In our model, the transient coupling boosting allows the invasion of bAPs into the dendritic compartment only during neuronal refractoriness, forming a positive feedback loop between somatic firing events and dendritic $Ca^{2+}$ plasticity updates to improve the SNR. We anticipate that extending the model to multi-compartmental neuron will help reveal how our proposed transient coupling mechanism may contribute to dendritic plateau potentials, a possibility we explore further in ongoing modeling work.

Transient boosting in dendritic coupling is thought to depend on various ionic channels that support the mechanistic transmission and regeneration of bAPs. Specifically, Na+ and K+ channels and their associated densities across the dendritic branches may underlie bAP transport to distal dendritic branches [58,62,63]. In principle, we may experimentally test the transient change of the active component of the coupling conductance by looking at the partial derivative of the membrane resistance along the proximal dendrites of PCs. However, measuring a rapid change in the axial resistance and its effect on the transitory part of signal transmission is technically difficult. On the other hand, plateau potentials only portray a snapshot of strong dendritic excitability which may correlate with somatic spiking but the causes underlying its gradual growth requires further testing.

On the other hand, an active component in dendritic branches might function similarly to our transient boosting mechanism, potentially enabling the generation of dendritic spikelets that co-occur with somatic spikes as a self-supervisory signal. We hypothesize that such an active component, exhibiting similar rise and decay dynamics, may indeed exist; however, its precise role requires further investigation.

### Underlying molecular mechanisms of our calcium-based learning rule

In the two compartmental models presented, our excitability signal $e(t) = C(t) - \overline{C(t)}$ (Eq 5), which measures the necessary amount of plasticity induction, depends on the long-term average of the intracellular calcium concentrations. Although we did not explicitly model this averaging process, it may involve interactions between free $Ca^{2+}$ ions and calmodulin-dependent protein kinases such as CaMKII and CaN, which are known to affect LTP and LTD, respectively [64–66] (see Yasuda et al. [67] for a recent review). In particular, CaMKII likely acts as a leaky integrator of $Ca^{2+}$ during LTP induction but not during maintenance of synapses [68]. Our model employed the multiplicative bounding function (Eq 17) to maintain the learned synaptic representations by suppressing a further LTP induction. Importantly, CaMKII binding rates are fast and this kinase acts as soon as $Ca^{2+}$ ions flow into the dendritic spine [69]. We conveniently chose an appropriate decay constant for $\overline{C(t)}$ that was sufficient for the task, but whether this constant matches the decay constant of intracellular kinase messaging remains to be understood.

The spike-trace-based plasticity rule relies on the artificially generated graded spike trace $Y(t)$, but biological cells do not have a readily available trace that informs spiking activity. With this and the vast amount of existing literature on calcium-based synaptic plasticity in mind (e.g., Inglebert et al. [45]), we replaced the artificial trace with the calcium concentration trace $C(t)$ in our calcium-based plasticity rule. Thus obtained calcium-based plasticity rule yielded a symmetric spike-timing rule similar to the Shaffer collateral STDP for spike singlet-doublet pairings. This contrasts with the predictions of a previous calcium-based STDP model for spike singlets, in which different levels of calcium entry result in stages

of LTD after a given $Ca^{2+}$ concentration threshold and those of LTP after at higher levels of $Ca^{2+}$ concentration [70]. It is noted that the Shaffer collateral also exhibits a symmetric STDP under dopaminergic modulation [71]. We consider our plasticity rule to be a generalized account of the course of synaptic plasticity signals with no strict requirements regarding threshold constraints or LTD/LTP amplitude specifications.

### Burst-triggered symmetric STDP in the calcium-based model

The proposed calcium-based plasticity rule produces synaptic strength changes resembling those observed in the burst-triggered STDP at the CA3-CA1 Schaffer collateral [23,56]. The results are interesting as bursts are thought to transmit spikes more faithfully than single spikes [15,72,73]. On the other hand, our model (and the experiment as well [56]: see Fig 5A) did not produce an asymmetric STDP time window reported in other studies for singlet-spike paring at the Shaffer collateral [20,21]. It was speculated that this discrepancy was partly due to differences in an experimental setting [56]: while a potassium-based solution induces LTD and hence gives asymmetric STDP, a cesium-based intracellular solution blocks potassium channels and depolarizes the postsynaptic neuron to enhance bAPs, which may turn LTD into LTP to generate symmetric STDP [21].

The Schaffer-collateral STDP only exhibits LTP for a small number of pairings (see Fig 5B). However, for a large number of singlet-doublet pairings, strong LTD components appear at both sides of the LTP time window (Fig 5C). The Schaffer-collateral STDP is also LTD dominant in singlet-singlet pairings (Fig 5A). Our neuron model claims that the LTP components for single-doublet pairing arise from a reliable bAP propagation (i.e., a sufficient invasion into dendritic branches). However, our model cannot account for the LTD components. A likely reason for the lack of LTD is that our STDP assessment ignores the presence of inhibition, which was suggested to induce LTD for a causal pre-post spike pairing [21,22]. In addition, our model does not include $Ca^{2+}$-dependent $K^+$ channels, which are known to halt positive loops mediated by $Ca^{2+}$ signaling [74–76]. These channels suppress the activation of CaMKII in spines [77], a kinase crucial for the LTP induction, and decrease intraburst firing rates, which may induce LTD [78]. All these possibilities have yet to be addressed experimentally.

### The role of pre-existing cell assemblies in rapid pattern detection

The ability of the brain to rapidly learn and remember features of sensory experiences has long been thought to require the experience-dependent generation of cell assemblies. However, whether and to what extent the learning of cognitive experiences is constrained by pre-existing cell assemblies in the brain has been debated [28–30,79]. Our network modeling results suggest that the proposed learning rule rapidly associates pre-existing assemblies with a salient pattern when within-assembly connections are sufficiently strong (Fig 6C, 6D).

Our model predicts that pre-existing assemblies are rapidly tuned to salient input patterns through a "Neighbor Supervision" mechanism. In this framework, some neurons within an assembly have a higher likelihood of spiking early in response to a given pattern. These early-spiking neurons act as supervisory signals, enhancing the excitability of other assembly members through recurrent excitation and facilitating recruitment during the tuning process. The stability of this rapid tuning is maintained by dynamic reorganization of recurrent inhibition (Fig 6E).

Experimentally, this mechanism could be tested by tracking spiking activity in cell assemblies during pattern presentation, identifying early-spiking neurons, and assessing how changes in excitatory or inhibitory synaptic strengths affect the speed and precision of tuning. Our findings suggest that strongly coupled, competing pre-existing assemblies provide the cortical machinery for rapid learning of activity patterns.

In machine learning-based approaches, competition among neurons during training can prevent overfitting by encouraging simpler and more general representations [80]. In the brain, assemblies of tightly connected neurons can compete for simple and robust representations of information carried by stimuli. As the identity of the assembly may change over time, which neuronal assembly gets recruited and wins over the rest may not matter [81,82].

## Limitation of the present models

The dendrite of our two-compartmental neuron model likely describes basal dendritic branches receiving sensory input rather than apical dendritic branches receiving top-down input. The nature of our Hebbian learning rule relies on strong somato-dendritic correlations during spiking activity, which is reminiscent of strong correlations between basal dendritic $Ca^{2+}$ events and somatic firing events in PCs. In contrast, evidence indicates that the correlation between $Ca^{2+}$ events and somatic spikes are weak in apical tuft dendrites [44,83] (but see Beaulieu-Laroche et al. (2019) [59]). Indeed, the strengthening of synapses highly depends on dendritic location [2,84,85]. Then, the question arises about how cortical neurons integrate bottom-up sensory information and top-down context information arriving at different dendritic locations [34,86]. The present model falls short for addressing this question. Exploring such integration requires more realistic separate compartmentalization [15,60,61] combined with compartment-specific plasticity mechanisms [2,85,87].

Our simplified models did not incorporate several ionic channels that can contribute to synaptic plasticity. With rising levels of $Ca^{2+}$, SK channels are known to terminate NMDAR activation via decreasing the local potential [75]. These channels may exert a significant effect on longer-lasting plasticity, and a future model needs to address this effect in a realistic morphological model with localized spines and intracellular $Ca^{2+}$ pools.

Given the non-linearity of intracellular processes, we suspect that translating the $Ca^{2+}$ transients into spatially localized spiny compartments across dendritic branches will likely require model refinements. In particular, appropriate channel densities might be required to address bAP invasion in the dendritic tree and subsequent synaptic plasticity cascading in its branches [41,42]. Proper bAP propagation is crucial for "self-supervising" the unsupervised learning proposed in this study. Addressing these limitations will require multi-compartmental neuron models, which provide a mechanistic investigation of our proposed transient boosting of dendritic coupling with somatic spiking. In addition to this, presynaptic axonal projections are known to make several postsynaptic contacts and the effect on spines, known to be highly effective at filtering out dendritic membrane potentials [88], remain to be explored in this context.

In sum, we have modeled the neurobiological mechanism underlying a rapid adaptation of pyramidal cell spiking responses to a statistically salient pattern hidden in the barrage of synaptic input. Our modeling work predicts the elements of single-cell and network-level computations, including a transiently boosted somato-dendritic coupling and pre-existing cell assemblies, crucial for this learning. The proposed mechanism may signal dendritic spikes or the boosting of dendritic potentials, and may underlie one-shot learning of episodes, a great advantage of the brain's memory systems.

## Methods

The following sections describe model equations in separate subsections. All parameter values discussed in these subsections are shown in S1 Table and S3 Table for presented models.

## Synaptic and intrinsic parameter selection

All synaptic time constants and conductance values were selected within biologically plausible ranges based on experimental data from hippocampal or cortical pyramidal neurons. Our goal was not to fit precise biophysical measurements but to capture realistic dynamics while maintaining computational tractability. Parameters were drawn from experimental studies where available, and in some cases adjusted to account for physiological temperature (e.g. NMDA time constant parameters as in Sáray et al. 2021 [50]). Parameter ranges are consistent with values reported in the literature (e.g., Spruston et al. 1995 [89], Bartos et al., 2002 [90]); see also curated compilations such as CompNeuro Waterloo, accessed June 2025) and original studies cited where appropriate. In particular, the model's qualitative behavior, including pattern-specific tuning, was robust across modest variations in these parameters provided that E/I balance is maintained.

## Selectivity index and tuning rapidity

Tuning rapidity is assessed by the selectivity index for each pattern and a threshold of activity. The selectivity index is computed across time following Eq 6:

$$s.i.(t) = tanh\left(\alpha.\frac{\mu_{w(pattern)}(t) - \mu_{w(non-pattern)}(t)}{\sigma_{w(non-pattern)}(t) + \epsilon}\right) \tag{6}$$

With $\epsilon = 10^{-6}$ providing numerical stability, and $\alpha = 0.5$ scaling the output for high selectivity when bursting to a pattern occurs. The index is normalized into the [-1, 1] range, and with this in mind we set a dynamic threshold $T$ of:

$$T = \mu_{w(non-pattern)}(t) + k.\sigma_{w(non-pattern)}(t) \tag{7}$$

With $k = 5$ (i.e. $k\sigma$ for 99.99994% confidence), the tuning rapidity to a given pattern in seconds is determined by the condition: $s.i.(t) > T$ and $s.i.(t) > 0.15$ (a lower bound). The tuning selectivity has to be maintained for at least 1 second of the simulation trial to consider the time when the condition was met.

## Initial bias sensitivity analysis

We assessed whether initial synaptic weights contribute to bias in pattern detection by analyzing the distribution of median initial weights for each input pattern across 100 simulations, separated by whether the pattern was eventually detected or not. Patterns were generated with parameters $r = 5$ Hz, $T_{pat} = 100$ ms, and no predefined bursting units (i.e. as in S1A Fig). To ensure fair comparisons, we selected the bounds of the Uniform distribution as [–3,3], which matches the approximate 6-$\sigma$(99.7%) support range of the Gaussian. For the LogNormal distribution, we used parameter values $\mu = -0.3466$, $\sigma = 0.8326$, yielding a distribution with approximately the same variance as the Gaussian (i.e., $\sigma \approx 1$). As LogNormal is strictly positive, inhibitory synaptic weights were just multiplied by -1 to make the overall mean zero. Mann–Whitney U test yielded statistically significant results for all initial conditions ($p < 10^{-4}$).

## Somatic compartment (spike trace-based)

The somatic membrane potential $V_s$ integrates input coming from the dendrite through the conductance $g_{cds}$ and leaks out through the conductance $g_{ls}$. $V_{ls}$ represents the leakage potential of soma, $C_s$ is the capacitance of the somatic compartment, and $V_d$ represents the dendritic membrane potential. The potential evolves following Eq 8:

$$\frac{dV_s}{dt} = \frac{g_{cds}(V_d - V_s) + g_{ls}(V_{ls} - V_s)}{C_s} \tag{8}$$

When $V_s > V_{th}$, a spike event occurs and the somatic membrane potential is updated to a reset voltage $V_s \rightarrow V_{re}$ where it remains refractory for $\tau_{ref}$ between 2 to 3 milliseconds (in all models). During a refractory period, the integration of Eq 8 is halted, and resumed afterward.

## Dendritic compartment (spike trace-based)

The dendritic membrane potential $V_d$ receives excitatory and inhibitory input through conductance-based currents. The dendritic potential evolves following

$$\frac{dV_d}{dt} = \frac{g_{ld}(V_{ld} - V_d) + g_e(V_{er} - V_d) + g_i(V_{ir} - V_d) + g_{csd}(V_s - V_d)}{C_d} \tag{9}$$

where $g_{ld}$ and $V_{ld}$ refer to the conductance and reversal potential of the dendritic leakage current, respectively, $g_e$ and $V_{er}$ to the conductance and reversal potential of excitatory synaptic current, $g_i$ and $V_{ir}$ to the conductance and reversal potential of inhibitory synaptic current, $g_{csd}$ to the conductance parameter of the soma-to-dendrite electric coupling, and $C_d$ is the capacitance of the dendritic compartment. We model the excitatory and inhibitory conductances as double exponential functions with rise $\tau_{x,rise}$ and decay $\tau_{x,decay}$ time constants:

$$g_x(t) = \overline{g_x}\, F(x, t), \qquad x = e \text{ or } i \tag{10}$$

where $\overline{g_x}$ is the peak conductance value, and the time of the peak is represented as

$$t_{x,peak} = \frac{\tau_{x,decay}\tau_{x,rise}}{(\tau_{x,decay} - \tau_{x,rise})} \ln\left(\frac{\tau_{x,decay}}{\tau_{x,rise}}\right) \tag{11}$$

A double exponential function $F(x,t)$ evolves according to the following Eqs 12-14:

$$F(x, t) = (\xi_{x,decay} - \xi_{x,rise}) / \left( e^{-\frac{t_{x,peak}}{\tau_{x,decay}}} - e^{-\frac{t_{x,peak}}{\tau_{x,rise}}} \right) \tag{12}$$

$$\frac{d\xi_{x,decay}}{dt} = -\frac{\xi_{x,decay}}{\tau_{x,decay}} + \kappa \sum I(i, t) \tag{13}$$

$$\frac{d\xi_{x,rise}}{dt} = -\frac{\xi_{x,rise}}{\tau_{x,rise}} + \kappa \sum I(i, t) \tag{14}$$

with $\kappa$ being a homeostatic balancing parameter (see below). Note that $F(x,t)$ is normalized between 0 and 1. Synaptic input from presynaptic neuron $i$ at time $t$ is given as

$$I(i, t) = \sum_{t_i} w(i)\delta(t - t_i), \tag{15}$$

where $t_i$ refers to the spike times of the presynaptic neuron and $\delta(t)$ is the Dirac's delta function.

**Spike trace-based plasticity rule**

Our proposal is a multiplicative Hebbian synaptic plasticity rule. In Eq 1, the postsynaptic potential term consists of the magnitude of the afferent input $I(i, t)$ that decays over time with a time scale activation $\tau_p$:

$$\tau_p \frac{d}{dt}PSP(i, t) = I(i, t) - PSP(i, t) \tag{16}$$

The last term in Eq 1 consists of an Alpha function with free parameters $\phi_\zeta$ and $\tau_\zeta$ chosen to accommodate the growth of synapses without requiring explicit boundaries.

$$\zeta\left(|w(i)|\right) = \left(\frac{|w(i)| - \phi_\zeta}{\tau_\zeta}\right) \exp\left(-\frac{|w(i)| - \phi_\zeta - \tau_\zeta}{\tau_\zeta}\right). \tag{17}$$

At any given time, $PI(i,t)$ corresponds to the inducible plasticity for synapse $i$. The synaptic weight $w(i,t)$ evolves continuously over time according to the following coupled differential equations.

$$\frac{d\Delta w(i)}{dt} = \frac{1}{\tau_\Delta}\left(PI(i,t) - \Delta w(i)\right),$$
$$\frac{dw(i)}{dt} = \eta\Delta w(i). \tag{18}$$

where $\tau_\Delta$ is the time constant governing how fast the change in synaptic weight, $\Delta w(i)$, tracks the difference between $PI(i,t)$ and $\Delta w(i,t)$. $\eta$ is the learning rate controlling how fast changes in $\Delta w(i)$ are integrated into the synaptic weight $w(i)$.

Finally, we avoid plastic changes that change the nature of the synapse (excitatory to inhibitory and vice-versa) following Eq 19 (Dale's Law):

$$\Delta w(i) = \begin{cases} 0 & \text{if } sign(w(i) + \eta\Delta w(i)) \text{ not equal to } sign(w(i)) \\ \Delta w(i) & \text{otherwise} \end{cases} \tag{19}$$

## Somatic compartment (calcium-based)

In the extended calcium-based model, the somatic compartment reproduces the rapid upswing and decay of a biophysical action potential. When synaptic inputs depolarize the dendrite sufficiently for the soma to cross the spike threshold $V_{th}$, the somatic membrane potential $V_s$ jumps within a single timestep to the peak voltage $V_{peak}$. At this moment, the somatic dynamics switch from Eq. 20 to Eq. 21 and, during the refractory period, the soma no longer integrates synaptic input, while the dendrite remains coupled and continues to receive a driving current from the elevated somatic voltage. Outside the refractory period, $V_s$ evolves according to

$$\frac{dV_s}{dt} = \frac{g_{cds}(V_d - V_s) + g_{ls}(V_{ls} - V_s)}{C_s}, \tag{20}$$

where $g_{ls}$ is the somatic leak conductance and $g_{cds}$ couples the dendrite to the soma. During the refractory period, the somatic leak conductance transiently increases ($g_{lsr} \gg g_{ls}$), yielding rapid hyperpolarization:

$$\frac{dV_s}{dt} = \frac{g_{lsr}(V_{ls} - V_s)}{C_s}. \tag{21}$$

This dynamic produces a brief interval in which the difference $V_s - V_d$ becomes positive. Concurrently, dendritic conductances $g_{csd}(t)$ receive spike-triggered inputs $m(t)$, generating a wave-like current into the dendrite. Crucially, once the soma has spiked, its voltage is no longer directly affected by synaptic inputs until the refractory period ends, ensuring a clean separation between spike generation and dendritic integration during this interval. The dendrite continues to evolve according to its own Equation which is dependent on the somatic membrane potential through the somato-dendritic coupling current $I_{csd}$.

## Dendritic compartment (calcium-based)

We include the calcium current description based on the dynamical rates obtained with Hodgkin-Huxley steady-state activation variables and forward-backward rate constants ($\alpha$ and $\beta$ respectively) for HVA-Ca$^{2+}$ current [43]. The rate constants depend on dendritic voltage $V_d$ and are measured at all time steps following the sets shown in Eqs 22–23.

$$m\alpha = \frac{0.055\left(-27 - V_d\right)}{e^{\frac{-27-V_d}{3.8}} - 1},$$

$$m\beta = 0.94 \, e^{\frac{(-75-V_d)}{17}}, \tag{22}$$

$$m_\infty = \frac{m\alpha}{m\alpha + m\beta},$$

$$\tau_m = \frac{1}{m\alpha + m\beta},$$

$$h\alpha = 0.000457 \, e^{\frac{-13-V_d}{50}},$$

$$h\beta = \frac{0.0065}{e^{\frac{(-V_d-15)}{28}} + 1}, \tag{23}$$

$$h_\infty = \frac{h\alpha}{h\alpha + h\beta},$$

$$\tau_h = \frac{1}{h\alpha + h\beta},$$

with parameters $m_\infty$, $h_\infty$, $\tau_m$, and $\tau_h$ defined dynamically, the activation and deactivation variables are integrated over time using Eq 24 and Eq 25, respectively.

$$\frac{dm}{dt} = \frac{m_\infty - m}{\tau_m}, \tag{24}$$

$$\frac{dh}{dt} = \frac{h_\infty - h}{\tau_h}. \tag{25}$$

Following the Ca$^{2+}$ current description in [43], we model this current with $m^2h$ kinetics as shown in Eq 26. In this equation, $g_{Ca}$ is the maximum calcium current conductance, and $V_{Ca}$ is the calcium reversal potential. Note that the defined direction of the calcium is inwards, i.e., depolarizing the compartment.

$$I_{Ca} = \overline{g_{Ca}} m^2 h (V_{Ca} - V_d). \tag{26}$$

Thus, the dendritic compartment evolves its membrane potential using the following Eq 27. $I_{csd}$ is the somato-dendritic coupling current which will be discussed below:

$$\frac{dV_d}{dt} = \frac{I_{csd} + g_e(V_{er} - V_d) + g_i(V_{ir} - V_d) + g_{ld}(V_{ld} - V_d) + g_{Ca} m^2 h (V_{Ca} - V_d)}{C_d} \tag{27}$$

**Somato-dendritic coupling current (calcium-based)**

To approximate the active propagation of somatic spikes into the dendrites observed in cortical pyramidal neurons, we implemented a transient increase in the somato-dendritic coupling $g_{csd}$ conductance during bAP events. This models the temporary enhancement of electrical coupling mediated by voltage-gated sodium and calcium currents in the apical trunk, which is known to facilitate dendritic calcium influx during somatic firing [41,42]. To start with, in biological cells, conductance between dendrites and the soma of the cell have both active and passive properties, and the active properties rely on the locality of the channels that are presently opened or closed. Since inputs that traverse the dendritic branches during non-refractory states propagate towards the soma, the coupling conductance is fairly constant dominated by its passive properties. On the other hand, when the neuron is engaged in firing an action potential –and thus entering the refractory state– this coupling is dominated by active properties as the soma experiences a diverse opening and closing of channels that repolarize the cell. We emulate the functional outcome of bAP-driven calcium entry by increasing the coupling conductance transiently during somatic spikes as shown in Eq 28. This simplification, in our two-compartmental

model, captures the momentary reduction in axial resistance associated with active conductances engaged during back-propagation [62,91].

$$I_{csd} = \begin{cases} g_{csdr}(V_s - V_d) & \text{during non-refractory states} \\ g_{csd}(V_s - V_d) & \text{during refractory states} \end{cases} \tag{28}$$

The transient increase of the coupling $g_{csd}$ occurs on every spike. Similarly to $g_e$ and $g_i$, the coupling conductance evolves over time following the Eqs 10–11. The parameters for this coupling are $\tau_{csd,rise}$, and $\tau_{csd,decay}$ for the rise and decay of the double exponential function. $g_{csdr}$ as the resting coupling value, and $m_{csd}$ as the amplitude of the increase in nS. On each spike, the exponentials $\xi_{csd,decay}$ and $\xi_{csd,rise}$ receive $m_{csd}$ as input (bAP strength/pulse) at $t = t_f$, where $t_f$ is the somatic spike time. The set of Eqs 29–30 describe the transient increase of coupling $g_{csd}$.

$$g_{csd} = g_{csdr} + F_{csd}(t), \tag{29}$$

$$F_{csd}(t) = (\xi_{csd,decay} - \xi_{csd,rise})/\left( e^{-\frac{t_{csd,peak}}{\tau_{csd,decay}}} - e^{-\frac{t_{csd,peak}}{\tau_{csd,rise}}} \right)$$

$$\frac{d\xi_{csd,decay}}{dt} = -\frac{\xi_{csd,decay}}{\tau_{csd,decay}} + m_{csd}(t) \tag{30}$$

$$\frac{d\xi_{csd,rise}}{dt} = -\frac{\xi_{csd,rise}}{\tau_{csd,rise}} + m_{csd}(t)$$

### NMDAR dynamics (Ca$^{2+}$/NMDAR-based)

Our last model includes NMDAR dynamics based on the set of Eqs 10-14 and a conductance-based current obtained from the following:

$$I_{NMDA} = g_{NMDA}.\,MgB(V_d)(V_{NMDA} - V_d)(1 - f), \tag{31}$$

$$I_{Ca} = g_{NMDA}.\,MgB(V_d)(V_{NMDA} - V_d)(f),$$

with $f$ a constant factor mediating NMDA-dependent calcium current influx $I_{Ca}$. The latter adds up to the HVA-Ca$^{2+}$ current described previously in Eq 26. Magnesium-block voltage-dependence function was taken from Jahr & Stevens (1990) [49]:

$$MgB(V) = \frac{1}{1 + \frac{[Mg2+]}{3.57} e^{(-0.062*V)}}, \tag{32}$$

with Magnesium concentration $[Mg2+]$ in $mM$. Time constant parameters for NMDAR are originally temperature-corrected values (Q10). Importantly, we consider that around 75% of the excitatory synapses in the model also activate NMDAR dynamics. This input is thus affected by integrating $g_{NMDA}$ as other conductances (see Eqs 10–11).

### Synaptic rescaling

We introduce a synaptic renormalization mechanism to maintain a balanced input strength throughout the simulation. This is implemented via a scalar constant, $\kappa$, which multiplies all synaptic weights at the moment of their integration (See Eqs 13-14). Rather than modeling a biophysical homeostatic process with a defined biological timescale, $\kappa$ is updated periodically every $T_\kappa$ milliseconds to preserve the overall synaptic displacements $\sum |w_i|$ (See Eq 33 and Eq 34). This discrete-time renormalization does not involve exponential decay and serves purely as a computational renormalization procedure.

Across all simulations, $\kappa$ remained above 1.0.

$$\vartheta^* = \sum_{i}^{n} |w(i)|, \tag{33}$$

$$\kappa\left(1 - \frac{\kappa\vartheta^* - \kappa\vartheta}{\kappa\vartheta}\right) \to \kappa \quad \text{at every } T_\kappa \text{ milliseconds.} \tag{34}$$

After update of the homeostatic parameter $\kappa$, we also update the sum of displacements $\vartheta \leftarrow \vartheta^*$.

### Few-shot learning in networks of pre-existing assemblies

Our network results include pre-existing assemblies with assembly sizes defined by a Gaussian distribution of mean = $\mu$ = 18 and standard deviation $\sigma = 3$. The cells in these networks had parameters as in Spike trace-based model from th different capacitance for excitatory and inhibitory cells to increase the firing rate of inhibitory units. The rest of the parameters and their values are depicted in S3 Table. In our analysis regarding weakly and strongly coupled assemblies (Fig 6D), we picked the six strongest activated assemblies across the trial simulation for given salient patterns on each network. We also conditioned the response, in terms of spike count relative to assembly size, of these assemblies to be above 1. Any assembly that spiked at least a number of spikes equal to the assembly size at the last given pattern presentation is thus considered in the analysis. This favors the weakly coupled pre-existing assemblies as their activity is much lower than in the strongly coupled case. We consider that this is appropriate as small groups (2-3 cells) within weakly coupled assemblies may be responsive to a given salient pattern but the assembly itself has yet to develop a group response.

### Supporting information

**S1 Fig. Stereotypical patterns and example tuning trial cases.** A: Raster plots of three stereotypical patterns generated from Poisson processes. Plots show pattern examples of 100 units spiking for $T_{pat}$ milliseconds at rate $r$. Pattern parameters are expressed in text on the top row. Bottom row are raster plots with 20% of units bursting. Bursting units are picked randomly. B: Selectivity index computed across time for an example random neuron simulation trial. The index of each pattern changes across time and the tuning speed (in seconds; brown dashed line) is obtained when the activity reaches the threshold (gray dashed line). C: Median initial synaptic weight value for detected (tuned) and non-detected (no tuning) patterns at different initial sampling distribution. Gaussian distributed synaptic weight values yield the least amount of bias. D: Simulation trial example showing, on the right, the dendritic and somatic compartment membrane potential traces (purple, and black respectively; top row), and somatic peristimulus time histogram (bottom) showing the spike response of the model to the learned pattern (blue). On the right, the selectivity index computed across time for the given trial, and tuning speed (in seconds) is obtained when conditions are met. E: Same as in D, but the input pattern now has 10% of its units bursting across time, decreasing the time it takes to develop a robust response (tuning time). (TIFF)

**S2 Fig. Kernel density estimation of excitatory and inhibitory synaptic weight $w$ distributions for red tuned trial (both models yield same results).** Colored lines from estimations at $t = 0, 6, 12, 20$ seconds of a single trial. Top row: Distributions for excitatory synapses pooled for red, green, and blue patterns. Across time, red pattern pooled excitatory synapses become bimodal with a large amount of strong synapses. Bottom row: Distributions for inhibitory synapses pooled for red, green, and blue patterns. Across time, blue and green pooled inhibitory synapses become bimodal exhibiting strong inhibition with blue and green pattern presentations relative to red pooled inhibitory synapses. Selectivity to a

single pattern is hinted as both dense excitatory links for red pattern as well as dense inhibitory links for green and blue pooled synapses.
(TIFF)

**S3 Fig. Example time course of conductance traces during pattern presentation for spike trace-based and calcium-based models.** The corresponding model layout on the left of each time course plot. Red pattern tuned trial. A: Spike trace-based model conductances (blue: excitatory, red: inhibitory, green: somato-dendritic) and dendritic membrane trace (black; axis on the right) during red pattern presentation at the end of the simulation. Vertical lines on top represent model output spikes. Inhibitory conductance plummets as an effect of learning while excitatory conductance remains high during red pattern presentation. B: Same as above but for green and blue pattern presentations. Inhibitory conductance is high relative to excitatory conductance. C: Same as in A but for calcium-based model conductances. $g_{csd}$ is transiently boosted on each somatic spike to enhance dendritic membrane potential peaks. D: Same as B but for the calcium-based model.
(TIFF)

**S4 Fig. Example trial simulation for strong LTD induction to non-related inputs for calcium-based variant rule during pattern tuning.** First panel: Membrane potential traces (left y-axis) of somatic (purple) and dendritic (gray) compartments. Second panel: Calcium $C(t)$ (blue) and low-pass filtered $\overline{C(t)}$ (midnight blue) traces. Third panel: Plasticity induction error signal $e(t)$. Fourth panel: Synaptic weight changes $\Delta w$ of input IDs #109 (scarlet) and #19 (pink) traces across time. Vertical marker indicating spike time of input. Strong potentiation is followed by a long-lasting LTD phase yielding a negative phase of $e(t)$ after the preferred pattern. Note that spikes of ID #19 (pink) occurring during the blue pattern do not affect plasticity induction of $w_{19}$ as the sample of $e(t)$ at given spike times is low (trend remains the same).
(TIFF)

**S5 Fig. Pairwise spike correlation matrix for the full network over the 10-second simulation.** Each entry represents the Pearson correlation between the binned spike trains of two neurons (25 ms bin width). Distinct blocks of elevated correlation emerge along the diagonal, corresponding to the predefined assemblies and confirming temporally coordinated spiking within these groups. Low correlations outside these blocks reflect weak synchrony between unrelated neurons. The white line displays neurons which exhibited no spiking activity during the simulation, leading to undefined correlations (NaN). The matrix demonstrates that structured intra-assembly connectivity results in reliably clustered spiking activity at the network level.
(TIFF)

**S1 Table. Membrane and synaptic parameter values across models.**
(PDF)

**S2 Table. Synaptic plasticity rule parameter symbols, values, and descriptions.**
(PDF)

**S3 Table. Recurrent network with assemblies: parameter symbols, values, and descriptions.**
(PDF)

# Acknowledgments

The authors are grateful to Yukiko Goda and Bernd Kuhn for the valuable discussion on the physiological properties of pyramidal neurons. We are also grateful to Milena M. Carvalho and Toshitake Asabuki for their helpful discussions and support.

## Author contributions

**Conceptualization:** Gaston Sivori, Tomoki Fukai.

**Investigation:** Gaston Sivori.

**Supervision:** Tomoki Fukai.

**Writing – original draft:** Gaston Sivori.

**Writing – review & editing:** Tomoki Fukai.

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
