## [Decision Letter · Decision Letter 0]

30 Apr 2025

PCOMPBIOL-D-25-00218

Transient Boosting of Action Potential Backpropagation for Few-shot Temporal Pattern Learning

PLOS Computational Biology

Dear Dr. Fukai,

Thank you for submitting your manuscript to PLOS Computational Biology. After careful consideration, we feel that it has merit but does not fully meet PLOS Computational Biology's publication criteria as it currently stands. Therefore, we invite you to submit a revised version of the manuscript that addresses the points raised during the review process.

In particular, both reviewers point out that that the model has a number of assumptions that need to be both better motivated and examined to determine whether they are critical. They also have some constructive suggestions about further experiments you can perform to make your arguments more compelling.

Please submit your revised manuscript within 60 days Jun 30 2025 11:59PM. If you will need more time than this to complete your revisions, please reply to this message or contact the journal office at ploscompbiol@plos.org. Please include the following items when submitting your revised manuscript:

We look forward to receiving your revised manuscript.

Kind regards,

Abigail Morrison

Academic Editor

PLOS Computational Biology

Hugues Berry

Section Editor

PLOS Computational Biology

**Journal Requirements:**

At this stage, the following Authors/Authors require contributions: Gaston Sivori, and Tomoki Fukai. Please ensure that the full contributions of each author are acknowledged in the "Add/Edit/Remove Authors" section of our submission form.

4) Your manuscript is missing the following section headings: Abstract, and Introduction.  Please ensure all required sections are present and in the correct order. Make sure section heading levels are clearly indicated in the manuscript text, and limit sub-sections to 3 heading levels. An outline of the required sections can be consulted in our submission guidelines here:

5) Please upload all main figures as separate Figure files in .tif or .eps format. For more information about how to convert and format your figure files please see our guidelines: 

6) We notice that your supplementary Figures are included in the manuscript file. Please remove them and upload them with the file type 'Supporting Information'. Please ensure that each Supporting Information file has a legend listed in the manuscript after the references list.

7) Please amend your detailed Financial Disclosure statement. This is published with the article. It must therefore be completed in full sentences and contain the exact wording you wish to be published.

1) If the funders had no role in your study, please state: "The funders had no role in study design, data collection and analysis, decision to publish, or preparation of the manuscript."

8) Please provide a completed 'Competing Interests' statement, including any COIs declared by your co-authors. If you have no competing interests to declare, please state "The authors have declared that no competing interests exist". Otherwise please declare all competing interests beginning with the statement "I have read the journal's policy and the authors of this manuscript have the following competing interests:"

**Reviewers' comments:**

Reviewer's Responses to Questions

**Comments to the Authors:**

**Please note that two reviews are uploaded as attachments.**

Reviewer #1: Review on:

Transient Boosting of Action Potential Backpropagation for Few-shot Temporal Pattern Learning

Summary:

The paper introduces a self-supervised plasticity rule for few-shot pattern learning. The plasticity rule consists of three parts in a two-compartment neuron model: the self-supervised error signal resembles the sliding window of BCM-rule except with eligibility trace of spike trains; the post-synaptic potential is eligibility trace of total input; the scaling term is a boundary factor for synaptic weight.

The original form is extended to a calcium-based form where the concentration of calcium serves as the substrate for eligibility traces and the error signal is mediated by bAP, i.e., somato-dendritic coupling. The calcium-based form is biologically plausible, and the simulations fit with single-doublet pairing protocol induced Schaffer-collateral STDP in hippocampus. The performance of pattern learning is further improved by the insertion of NMDARs which promotes bursting activities.

Lastly, the paper proposes a recurrent scheme with preconfigured small assemblies. The original form efficiently learns the input pattern with just several shots by recruiting the pre-existing assemblies.

The paper in general did solid modeling and simulations. The idea of using calcium as the substrate for eligibility trace is interesting. However, there are some subtle assumptions that need further checking, especially the motivation for the complexity of models. Next part are the major comments regarding these assumptions and model choice. In the last part, I have some minor comment line by line.

Major comments:

1. In all settings, the feedforward(input) connections are all-to-all. This would nevertheless induce some level of initial bias in network response, e.g., some neurons would have higher response to certain input patterns. How strong is this initial bias in your model? In addition, how sensitive is performance to initialization of weights? (e.g., instead of gaussian, log-normal or uniform)

2. The homeostasis has extremely fast updating (time constant of 1.0ms), this does not align with experiments that homeostasis is usually of a much longer time scale.

3. In eq.18, an extra low-pass filter (time constant 100ms) is used for PI(i,t). Is this necessary as PI(i,t) itself is the product of filtered signals e(t) and PSP(i,t)?

4. In the Ca-NMDARs model, the AMPA/NMDA ratio is chosen as 5%. Is there experimental support or just to have minimal effect on dend depolarization? In addition, the effect of NMDARs is to promote bursting. What is the difference between NMDARs insertion and explicitly setting 10% input neurons to burst?

5. There is no mentioning of synaptic delay in either the single neuron or network model. How is it defined? Would synaptic delay change your results?

6. In the recurrent network model, the connectivity is set to a scheme where inhibition is strongly recruited by excitatory neurons (Exc-Inh is very strong). It looks like a trivial choice. Is this a necessary condition?

7. The authors argue that the pre-existing assemblies provide initially active neurons. What about the pre-existing within-assembly connectivity? Would random connectivity with some neurons being intrinsically more active give similar results?

8. A general concern is there are so many parameters hence different combinations could lead to the same results(redundancy). How do you argue for the choice of conductance and time constants? More, specifically how do you justify the relative value of different conductance and time constants?

Detailed comments:

1) Page 6 Eq. 17, the Alpha function gives rise to typical spine head volume distribution under what condition?

2) Page 6 bottom lines, why equal number of exc and inh input neurons?

3) Page 7 paragraph 2 last sentence, what is the distribution of early and late onset responses?

4) Page 8 paragraph 1, how is the long-tailed strength distribution related to bimodality of trained weights?

5) Fig1.F, e(t) is always on(non-zero). What happens if there is a period of silence? Would it drastically change weights?

6) Page 10 paragraph 2 last line, LTP and LTD become equivalent. Is it strictly or roughly equivalent?

7) Page 13 paragraph 2, what is opposite conductance of the somato-dendritic coupling?

8) Page 13 second-last line, is it “(Supp Fig. 3B and 3D)”?

9) Fig.2D, add units to color bar.

10) Fig.3B, the confidence interval of mean estimation? In Fig2.C, the variance seems very large, therefore is the mean estimation meaningful?

11) Page 18 paragraph 2, what is the use of synaptic transmission failure? Simply adding more noise to the Poisson spike trains?

12) Page 28 paragraph 1 last line, with or without?

13) Page 28 paragraph 2, last line, be hinted at?

14) Fig.6B, the raster plot is a bit messy. What does pair-wise correlation look like?

15) Fig.6E, does the normalized spike count increase because of decrease of assembly size?

16) Page 3 paragraph 2, (but see) what?

17) Eq 31, is it Ica (+=) or a different variable Ica+ from Eq 26?

Reviewer #2: My review is uploaded as an attachment

**Have the authors made all data and (if applicable) computational code underlying the findings in their manuscript fully available?**

Reviewer #1: Yes

Reviewer #2: Yes

PLOS authors have the option to publish the peer review history of their article (what does this mean?). If published, this will include your full peer review and any attached files.

Reviewer #1: **Yes: **Lihao Guo

Reviewer #2: No

**Figure resubmission:**
---

## [Decision Letter · Decision Letter 1]

1 Sep 2025

PCOMPBIOL-D-25-00218R1

Transient Boosting of Action Potential Backpropagation for Few-shot Temporal Pattern Learning

PLOS Computational Biology

Dear Dr. Fukai,

Thank you for submitting your manuscript to PLOS Computational Biology. After careful consideration, we feel that it has merit but does not fully meet PLOS Computational Biology's publication criteria as it currently stands. Therefore, we invite you to submit a revised version of the manuscript that addresses the points raised during the review process.

Reviewer 2 points out an important risk that the effects shown are dependent on unbiological model assumptions. It is critically important to resolve this issue for the paper to be published in PLOS CB.

Please submit your revised manuscript within 60 days Nov 01 2025 11:59PM. If you will need more time than this to complete your revisions, please reply to this message or contact the journal office at ploscompbiol@plos.org. Please include the following items when submitting your revised manuscript:

We look forward to receiving your revised manuscript.

Kind regards,

Abigail Morrison

Academic Editor

PLOS Computational Biology

Hugues Berry

Section Editor

PLOS Computational Biology

**Reviewers' comments:**

Reviewer's Responses to Questions

Reviewer #1: The main concerns are all answered and explained.

Reviewer #2: I thank the authors for providing their replies to my concerns. However, at the moment, I do not feel that they have been addressed adequately. Pertaining concerns (1-2), I still find the dynamic coupling mechanism to be poorly motivated. The cited references (62, 91) certainly pertain to the backpropagation of APs, but do not appear to suggest a dynamic somato-dendritic coupling conductance. I insist on this issue, because I think it may be crucial for the functioning of the model, and also related to my 3rd major concern about the high firing rates. In the parameter table, the spike threshold voltage is stated to be -50 mV. As the model is an I&F one, this means that the somatic compartment can not drive the dendritic compartment to a higher voltage; even at infinite coupling, the soma could only drive the dendrite to -50 mV at most. Conversely, this means a high number of spikes is required to elicit a sufficient dendritic calcium response. If the bAP would be modelled in a different way, e.g. through a spike-triggered dendritic current, the dendritic bAP amplitude could be tuned independently of the somatic threshold, leading to a calcium response that could be achieved for lower firing rates. As it stands, it appears that this strong firing rate increase is necessary to achieve a sufficient calcium response and eligibility signal (cf. Fig 2). It should also be noted that in a more physiologically plausible situation, potassium currents would shut off the calcium response as well as the high firing rates. Together, this makes me question whether the described few shot learning hinges crucially on this high firing rate, and whether it could be achieved under more plausible response characteristics. In their reply, the authors refer to using adaptation and a more plausible learning mechanism in future work to achieve the few-shot learning with more physiological response characteristics. My issues with this is that I can not assess this claim.

I therefore maintain my recommendation of major revision, and feel the authors should either point out the flaw in my reasoning or provide a better exploration of these different settings. This would also make the paper more generalisable.

**Have the authors made all data and (if applicable) computational code underlying the findings in their manuscript fully available?**

Reviewer #1: Yes

Reviewer #2: None

PLOS authors have the option to publish the peer review history of their article (what does this mean?). If published, this will include your full peer review and any attached files.

Reviewer #1: No

Reviewer #2: No

**Figure resubmission:**
---

## [Decision Letter · Decision Letter 2]

21 Nov 2025

Dear Dr. Fukai,

We are pleased to inform you that your manuscript 'Transient Boosting of Action Potential Backpropagation for Few-shot Temporal Pattern Learning' has been provisionally accepted for publication in PLOS Computational Biology.

Best regards,

Hugues Berry

Section Editor

PLOS Computational Biology

Hugues Berry

Section Editor

PLOS Computational Biology

Reviewer's Responses to Questions

**Comments to the Authors:**

Reviewer #2: With their reply to my comments and the changes to the model description in the manuscript, the authors have clarified how this critical model component works, where the somatic spiking can drive the dendritic voltage to sufficiently depolarized values to engage the Ca-dynamics. My recommendation is to integrate the figure from their reply into the manuscript to clarify this to the reader as well.

**Have the authors made all data and (if applicable) computational code underlying the findings in their manuscript fully available?**

Reviewer #2: Yes

PLOS authors have the option to publish the peer review history of their article (what does this mean?). If published, this will include your full peer review and any attached files.

Reviewer #2: No

---

## [Editor Report · Acceptance letter]

PCOMPBIOL-D-25-00218R2

Transient Boosting of Action Potential Backpropagation for Few-shot Temporal Pattern Learning

Dear Dr Fukai,

I am pleased to inform you that your manuscript has been formally accepted for publication in PLOS Computational Biology. Your manuscript is now with our production department and you will be notified of the publication date in due course.

With kind regards,

Zsofia Freund
